# HIERARCHICAL VALUE-DECOMPOSED OFFLINE REINFORCEMENT LEARNING FOR WHOLE-BODY CONTROL

**Zhilong Zhang**[1,2]*, **Yunpeng Mei**[3]*, **Xinghao Du**[1,2]*, **Hongjie Cao**[3]*, **Haonan Wang**[2],
**Pengyuan Min**[3], **Chenyu Wang**[3], **Pengfei Chen**[3], **Chenbo Xin**[3], **Yijie Wang**[3],
**Wenyu Luo**[2], **Yihao Sun**[4], **Yidi Wang**[1,2], **Lei Yuan**[1,2], **Gang Wang**[3]†, **Yang Yu**[1,2]†

[1]National Key Laboratory for Novel Software Technology, Nanjing University, China
[2]School of Artificial Intelligence, Nanjing University, China
[3]School of Automation, Beijing Institute of Technology, China
[4]Mila-Quebec AI Institute & Université de Montréal, Canada
`{zhangzl,duxh,wangyd}@lamda.nju.edu.cn`
`{231300049,231300065}@smail.nju.edu.cn`
`{meiyunpeng,hongjie.cao,mpy,wcy}@bit.edu.cn`
`{chenpengfei,xinchenbo,wangyijie,gangwang}@bit.edu.cn`
`yihao.sun@mila.quebec`
`{yuanl,yuy}@nju.edu.cn`

## ABSTRACT

Scaling imitation learning to high-DoF whole-body robots is fundamentally constrained by the scarcity of expert demonstrations. In contrast, large amounts of suboptimal data are readily available and offer a practical way to alleviate supervision bottlenecks in real-world whole-body control. However, leveraging such data introduces two central challenges: how to extract informative signals from imperfect trajectories, and how to cope with the increased learning complexity induced by high-dimensional control. To overcome this, we propose **HVD** (Hierarchical Value-Decomposed Offline Reinforcement Learning). The offline RL formulation provides principled data selection over suboptimal datasets, enabling the policy to prioritize high-value behaviors while down-weighting harmful ones. Complementarily, hierarchical value decomposition organizes learning along the robot's kinematic structure, improving credit assignment and reducing learning complexity in high-DoF systems. Built on a Transformer-based architecture, HVD supports *multi-modal* and *multi-task* learning, allowing flexible integration of diverse sensory inputs. To enable realistic evaluation and training, we further introduce **WB-50**, a 50-hour dataset of teleoperated and policy rollout trajectories annotated with rewards and preserving natural imperfections, including partial successes, corrections, and failures. Experiments show HVD significantly outperforms existing baselines in success rate across complex whole-body tasks. Our results suggest effective policy learning for high-DoF systems can emerge not from perfect demonstrations, but from structured learning over realistic, imperfect data. Our code is available at https://github.com/LAMDA-RL/HVD.

## 1 INTRODUCTION

Imitation learning has become a central paradigm for robotic policy learning, enabling robots to acquire complex skills directly from demonstrations (Pomerleau, 1991; Ross et al., 2011; Brantley et al., 2020; Feng et al., 2026). Recent advances such as diffusion based action generation (Chi et al., 2023; Ze et al., 2024; Liu et al., 2024) and vision language action models for instruction conditioned control (Kim et al., 2024; Black et al., 2024; Liu et al., 2024; Hu et al., 2024; Intelligence et al., 2025) further extend imitation learning toward unified perception, reasoning, and control systems, positioning it as a foundation for general purpose robotics.

---

*Equal contribution.
†Corresponding author.

However, scaling imitation learning to high DoF whole body robots exposes a fundamental bottleneck: expert data scarcity. Whole body control requires coordinated motion across many joints, dramatically enlarging the state and action spaces (Bellman, 1966; Kober et al., 2013). Collecting high quality demonstrations through teleoperation is cognitively demanding and physically costly, making large scale expert supervision impractical (Zhou et al., 2023; Jiang et al., 2025). In contrast, large amounts of suboptimal data naturally arise from teleoperation and policy rollouts. These trajectories often contain partial successes, corrections, and failures, and therefore provide a scalable but imperfect source of supervision.

Leveraging such data introduces two central challenges. First, suboptimal trajectories mix useful behaviors with harmful ones, requiring principled mechanisms to extract informative signals. Second, high DoF control increases learning complexity, making credit assignment difficult in long horizon and high dimensional settings. Existing imitation learning methods typically assume expert demonstrations, while standard offline reinforcement learning struggles to scale effectively to structured whole body systems with multi modal observations (Levine et al., 2020; Kalashnikov et al., 2018; Kumar et al., 2022).

To address these challenges, we propose **HVD** (Hierarchical Value Decomposed Offline Reinforcement Learning), a structured offline RL framework for whole body control. We first perform reward labeling over collected trajectories to provide consistent supervision signals across suboptimal data. Built upon these reward annotated datasets, HVD leverages offline RL for value based data selection, enabling the policy to prioritize high value behaviors while suppressing harmful ones. Meanwhile, hierarchical value decomposition organizes learning along the robot kinematic structure, improving credit assignment and reducing learning complexity in high DoF systems. Implemented with a Transformer backbone, HVD supports multi-modal and multi-task learning, making it more flexible.

To support realistic evaluation and training, we introduce **WB 50**, a 50 hour dataset of teleoperated and policy rollout trajectories annotated with rewards and preserving natural imperfections. Experiments on diverse whole body manipulation tasks show that HVD significantly outperforms existing baselines in success rate. Our results suggest that effective high DoF policy learning can emerge not from perfect demonstrations, but from structured learning over abundant imperfect data.

The primary contributions of this work are as follows:

- We propose HVD, an offline RL method for whole-body control via hierarchical Q-value decomposition with temporal chunking, enabling precise credit assignment in high-DoF, long-horizon tasks.
- We implement HVD using a Transformer-based architecture that supports multi-modal inputs and multi-task learning.
- We introduce WB-50, a 50-hour whole-body robotics dataset of imperfect, reward-labeled trajectories.
- We demonstrate that HVD outperforms baselines across diverse whole-body tasks and policy architectures, and validate the effectiveness of HVD on multi-task settings.

## 2 Preliminaries

**Markov Decision Process.** We model the robot control task as a Markov decision process (MDP) defined by the tuple $\mathcal{M} = (\mathcal{S}, \mathcal{A}, P, r, H)$, where $H$ is the horizon. The state space $\mathcal{S} = \mathcal{S}_{\text{obs}} \times \mathcal{S}_{\text{prop}}$ includes egocentric observations (e.g., images, point clouds) and proprioceptive state. The action space $\mathcal{A} \subset \mathbb{R}^d$ consists of joint-level commands for a $d$-DoF robot. and $r(s^h, a^h) \in [0, 1]$ is the reward function and $P(s^{h+1}|s^h, a^h)$ characterizes the non-stationary transition function of this MDP, which is a critical assumption because the whole-body movement and the constraints of the camera's field of view (FoV) result in partial observability, making the observation-based transitions appear highly stochastic and time-varying throughout the horizon. The goal is to learn a policy $\pi(a|s)$ that maximizes the expected return $V(\pi) = \mathbb{E}_\pi[\sum_{h=0}^{H} r(s_h, a_h)]$.

**Offline Reinforcement Learning** considers the problem of learning a policy from a fixed dataset $\mathcal{D} = \{(s, a, r, s')\}$ without further environment interaction. A central challenge is *distributional shift*, where the learned policy may query actions outside the support of the data, leading to erroneous value estimates and thus poor performance (Kumar et al., 2019; Levine et al., 2020; Koh et al., 2021).

**Implicit Diffusion Q-Learning (IDQL)** (Hansen-Estruch et al., 2023) builds on IQL Kostrikov et al. (2021), which can be viewed as an *actor-critic method* (Konda & Tsitsiklis, 1999), where the critic objective induces an implicit, behavior-regularized actor to prevent the value overestimation problem in offline RL. In this framework, the value function $V_\psi(s)$ is obtained by minimizing a convex loss over dataset actions:

$$V_\psi^*(s) = \min_\psi \ \mathbb{E}_{a \sim \mu(a|s)} \left[ f\big(Q_\theta(s^h, a^h) - V_\psi(s^h)\big) \right],$$

where $f$ is chosen as an asymmetric convex function (e.g., expectiles (Kostrikov et al., 2021), quantiles (Koenker & Hallock, 2001), or exponential (Beirlant et al., 1999)), determining how the implicit policy $\pi_\phi$ deviates from the behavior policy $\mu$. The Q-function is trained with Bellman backups:

$$\mathcal{L}_Q(\theta) = \mathbb{E}_{(s^h, a^h, s^{h+1}) \sim \mathcal{D}} \left[ \big(r(s^h, a^h) + V_\psi(s^{h+1}) - Q_\theta(s^h, a^h)\big)^2 \right],$$

To recover the policy, IDQL employs $\pi_\phi(a|s)$ trained via advantage weighted regression (Sasaki & Yamashina, 2020):

$$\mathcal{L}_\pi^{\text{weightbp}}(\phi) = \frac{1}{H} \sum_{h=1}^{H} \mathbb{E} \left[ \frac{|f'(Q(s^h, a^h) - V^*(s^h)|}{|Q(s^h, a^h) - V^*(s^h)|} \left\| \epsilon - \pi_\phi(\sqrt{\hat\alpha} a^h + \sqrt{1 - \hat\alpha}\epsilon, s^h, t) \right\| \right],$$

where $\epsilon \sim \mathcal{N}_{\text{pol}}(0, I)$ denotes Gaussian noise, $t$ is the noising timestep, $\hat\alpha_t$ is the noise schedule parameter in diffusion training, and $f' = \frac{\partial f}{\partial V(s)}$ denotes the derivative of $f$ with respect to $V(s)$.

## 3 Challenges in Whole-body Control

In this section, we argue that whole-body control presents two key challenges: the increased DoF and non-stationary observation dynamics. We examine the impact of this phenomenon on policy performance from both theoretical and empirical perspectives.

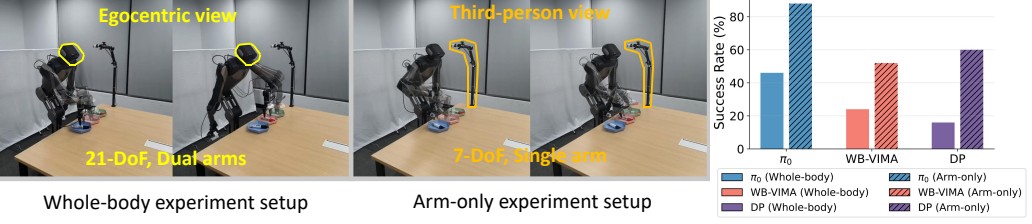

Whole-body experiment setup          Arm-only experiment setup

Figure 1: Performance comparison between 21-DoF whole-body policies and 7-DoF arm-only policies trained on 50 expert demonstrations.

### 3.1 Theoretical Analysis

In the context of **behavior cloning** (BC), theoretical studies have established that the *expert sample complexity*, defined as the number of expert trajectories required to learn a policy with a desired level of performance, scales poorly with the size of the policy set $\Pi$. This has been a focus of recent work (Rajaraman et al., 2020; Tu et al., 2022; Foster et al., 2024; Xu et al., 2024a). We begin by formally introducing the policy covering number of a policy class.

**Definition 3.1** (Policy covering number). For a policy class $\Pi \subset \{\pi_h : \mathcal{X} \to \Delta(\mathcal{A})\}^*$, we set that $\Pi' \subset \{\pi_h : \mathcal{X} \to \Delta(\mathcal{A})\}$ is an $\varepsilon$-cover if for all $\pi \in \Pi$, there exists $\pi' \in \Pi'$ such that for all $x \in \mathcal{X}$, $a \in \mathcal{A}$, and $h \in [H]$,

$$\log\left(\frac{\pi_h(a|x)}{\pi_h'(a|x)}\right) \le \varepsilon. \tag{1}$$

We denote the size of the smallest such cover by $\mathcal{N}_{\text{pol}}(\Pi, \varepsilon)$.

---

[*]While diffusion policies are typically implemented as implicit generative models, they theoretically induce an explicit probability density function $\pi(a|x)$ via the *probability flow ODE* formulation (Song et al., 2021). This bijective mapping ensures that the log-density $\log \pi(a|x)$ is well-defined and computable, rendering diffusion policies compatible with this definition.

We then formalize the dependence of the expert sample complexity of BC on this measure:

**Proposition 3.2** (Expert sample complexity of behavior cloning (Foster et al., 2024)). *For any expert policy $\pi^\star \in \Pi$, to ensure that the suboptimality gap of the learned policy $\hat{\pi}$ satisfies $V(\pi^\star) - V(\hat{\pi}) \leq \varepsilon$ with probability at least $1 - \delta$, the number of expert trajectories $n$ required for behavioral cloning needs to satisfy*

$$n = \tilde{\mathcal{O}}\left(\frac{H^3 \log \mathcal{N}_{pol}(\Pi, \varepsilon_\pi)}{\varepsilon^2}\right). \tag{2}$$

*Here, $H$ is the task horizon, $\varepsilon$ is the target suboptimality gap, and $\mathcal{N}_{pol}(\Pi, \varepsilon_\pi)$ denotes the $\varepsilon_\pi$-policy covering number of the policy class $\Pi$.*

This result reveals a critical quantitative limitation of BC: the expert sample complexity of BC is fundamentally tied to the $\log$-covering number of the policy class $\Pi$. As the **complexity or size of the action space expands**, $\mathcal{N}_{\mathbf{pol}}(\Pi, \varepsilon_\pi)$ **grows**, which in turn necessitates a larger number of expert trajectories $n$ to learn an $\varepsilon_\pi$-optimal policy.

### 3.2 EMPIRICAL VALIDATION

Figure 1 provides an empirical comparison of arm-only and whole-body policies applied to the same task. The leftmost and middle panels show the experimental setups for both cases: the whole-body setup uses a 21-DoF robot with dual arms and an egocentric view, while the arm-only setup uses a 7-DoF robot with a single arm and a third-person view. The rightmost panel compares the success rates of various policies ($\pi_0$, WB-VIMA, and DP) with the same number of expert demonstrations (50) for both the whole-body and arm-only setups. The whole-body policies consistently achieve significantly lower success rates than their arm-only counterparts across all models. This performance gap highlights the challenges posed by high-dimensional action spaces and non-stationary transitions.

In summary, the increased DoF in whole-body control leads to an explosion in expert data requirements, driven both by the complexity of DoFs and the inherent non-stationary transition.

## 4 METHOD

### 4.1 WB-50: IMPERFECT DATA FOR WHOLE-BODY RL

The preceding analysis shows that directly relying on expert demonstrations is impractical due to the increased DoF and instability of egocentric observations. Fortunately, in realistic settings, *non-expert demonstrations* are more abundant, arising naturally from teleoperation and policy rollouts (Zhou et al., 2023). To leverage this, we introduce **WB-50** as illustrated in Figure 2: a reward-labeled whole-body dataset spanning **over 50 hours**. WB-50 is intentionally composed of three distinct data sources to reflect realistic data distributions: i) **expert demonstrations** (43.7%), ii) **imperfect teleoperation** (14.6%), and iii) **policy rollouts** (41.7%) — the latter two comprising the majority, mirroring the scarcity of perfect supervision in practice. Every trajectory is annotated at the subtask level and labeled with discrete reward signals. More details are listed in Appendix B.

A direct way to leverage such data is offline RL, which enables policy learning from static and imperfect datasets (Lange et al., 2012; Levine et al., 2020). However, existing offline RL methods face fundamental limitations when applied to whole-body control. Most prior successes have been restricted to relatively low-dimensional tasks (Mandlekar et al., 2022; Sinha et al., 2022; Zhou et al., 2023), and current algorithms struggle to scale to the high degrees of freedom inherent in whole-body robots. Compounding this difficulty, whole-body control often involves sparse reward signals, which exacerbate the challenges of temporal credit assignment and policy optimization. Furthermore, prevailing approaches are typically confined to single-task or single-modality domains in embodied control, raising concerns about their versatility and scalability.

### 4.2 HVD: HIERARCHICAL VALUE-DECOMPOSED OFFLINE RL

To address the above issues, we introduce **H**ierarchical **V**alue-**D**ecomposed Offline Reinforcement Learning (HVD), designed for high-dimensional, whole-body control in robotic systems. Unlike conventional approaches that decompose the policy (Sentis & Khatib, 2006), HVD introduces hierarchy directly into the Q-value function through spatial decomposition. This key design allows

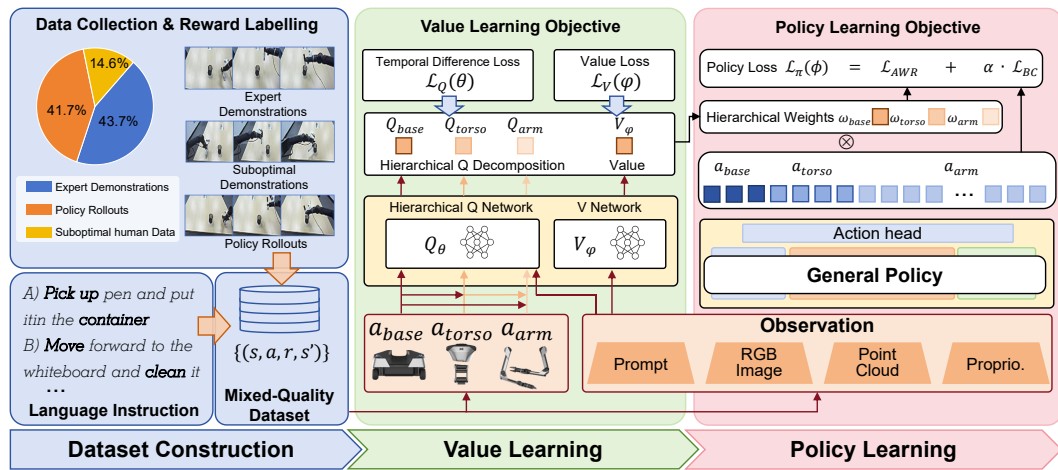

Figure 2: Framework of HVD. The proposed HVD framework consists of three stages: (1) Dataset construction and reward labeling from expert, suboptimal, and rollout data; (2) Hierarchical value learning with kinematically decomposed Q-functions and temporal chunking; and (3) Policy learning via hierarchical advantage-weighted regression. The diagram illustrates the data flow from multi-modal observations through hierarchical value function learning, and ultimately to policy training.

us to maintain a unified policy network while enabling fine-grained, component-specific value assessment for different parts of the robot. As a result, HVD achieves more accurate credit assignment across long-horizon, multi-step behaviors for whole-body control.

**Q-value Decomposition.** Inspired by Smith et al. (2012); Pan et al. (2024); Jiang et al. (2025), the action space of whole-body control policies can be decomposed into three hierarchical components corresponding to its physical structure, $\mathcal{A} = \mathcal{A}_{\text{base}} \times \mathcal{A}_{\text{torso}} \times \mathcal{A}_{\text{arm}}$. At each time step $h$, the action chunk of size $k$ is defined as $a^{h:h+k} = (a_{\text{base}}^{h:h+k}, a_{\text{torso}}^{h:h+k}, a_{\text{arm}}^{h:h+k})$, where each component represents a sub-action controlling a specific subset of the robot's degrees of freedom. Specifically, $a_{\text{base}}^{h:h+k}$ governs the lower-body motion (e.g., locomotion or base movement), $a_{\text{torso}}^{h:h+k}$ controls the upper body or torso orientation, and $a_{\text{arm}}^{h:h+k}$ manages the arm movements.

Based on this decomposition, we define hierarchical Q-values over temporal chunks of length $k$, where each level accumulates value estimates conditioned on progressively more complete subsets of the robot's action space. Specifically, for a chunk starting at timestep $h$, we compute:

$$\begin{cases} Q_{\text{base}}^{h:h+k} = Q_\theta(s^h, a_{\text{base}}^{h:h+k}), \\ Q_{\text{torso}}^{h:h+k} = Q_\theta(s^h, a_{\text{base}}^{h:h+k}, a_{\text{torso}}^{h:h+k}), \\ Q_{\text{arm}}^{h:h+k} = Q_\theta(s^h, a_{\text{base}}^{h:h+k}, a_{\text{torso}}^{h:h+k}, a_{\text{arm}}^{h:h+k}). \end{cases} \tag{3}$$

Here, each Q-value corresponds to a specific part of the robot, creating a layered structure for the value function, enabling more precise, joint-level credit assignment.

**Hierarchical Value Estimation.** To train the hierarchical Q-function in Equation 3, we employ a multi-level TD learning loss that aligns each partial Q-value with its corresponding estimated return:

$$\mathcal{L}_i^h(\theta) = \mathbb{E}\left[\left(r(s^h, a^{h:h+k}) + V_\psi(s^{h+k+1}) - Q_i^{h:h+k}\right)^2\right], \quad \text{with } i \in \{\text{base}, \text{torso}, \text{arm}\}, \tag{4}$$

where $r(s^h, a^{h:h+k}) = \sum_{j=h}^{h+k} r(s^j, a^j)$ is the reward for executing the action chunk $a^{h:h+k}$ on the state $s^h$, which aggregates the per-timestep rewards over the sub-episode from time $h$ to $h + k$. Moreover, $V_\psi(s^{h+k+1})$ represents the estimated value of the next state predicted by a value network parameterized by $\psi$. The Q-learning objective is designed to minimize the temporal difference between the predicted Q-value and the target value, and the total Q loss combines all hierarchical levels defined as below:

$$\mathcal{L}_Q(\theta) = \frac{1}{H} \sum_{h=1}^{H} \left[\mathcal{L}_{\text{base}}^h(\theta) + \mathcal{L}_{\text{torso}}^h(\theta) + \mathcal{L}_{\text{arm}}^h(\theta)\right]. \tag{5}$$

The result is a value decomposition that promotes credit assignment across space with temporal Q chunking, enabling more sample-efficient and coordinated whole-body control.

**Implicit Value Learning.** Concurrently, we incorporate implicit value learning to align value estimates across the hierarchical levels of our framework. By leveraging an in-sample learning paradigm (Wainwright, 2019; Hansen-Estruch et al., 2023), HVD effectively mitigates the risk of value overestimation caused by OOD actions, a persistent challenge in Q-learning methods:

$$\mathcal{L}_V(\psi) = \frac{1}{H}\sum_{h=1}^{H}\mathbb{E}\left[\sum_{i\in\{\text{base,torso,arm}\}}\left[\alpha\exp\left(Q_i^{h:h+k} - V_\psi(s^h)\right) - \alpha\left(Q_i^{h:h+k} - V_\psi(s^h)\right)\right]\right], \quad (6)$$

where $\alpha > 0$ is a temperature parameter controlling the strength of the constraint.

By optimizing this loss, we establish a soft lower bound on the value estimates across all hierarchical Q-heads, ensuring that limb-level value predictions remain aligned with global, whole-body goals.

**Policy Learning.** We train the policy network $\pi_\phi$ using a hierarchical variant of advantage-weighted regression (AWR) (Peters & Schaal, 2007; Peng et al., 2019; Nair et al., 2020). Rather than uniformly imitating all actions in the dataset, our method assigns importance weights to action chunks based on estimated advantages, encouraging the policy to prefer high-value behaviors while down-weighting low-return trajectories. This weighting is applied separately at each hierarchical level, allowing critical sub-actions to be emphasized even when other components generate lower returns.

The per-level advantage weight for an action chunk $a_i^{h:h+k}$ executed from state $s^h$ is defined as:

$$\omega_i^{h:h+k}(s^h, a_i^{h:h+k}) = \frac{\alpha\left|\exp\left(\alpha(Q_i^{h:h+k} - V_\psi(s^h))\right) - 1\right|}{|Q_i^{h:h+k} - V_\psi(s^h)|}, \quad (7)$$

where $\alpha > 0$ controls the sharpness of advantage-based reweighting.

This formulation ensures that actions with higher relative advantage receive exponentially increasing weight, while preserving gradient flow even near the decision boundary. Furthermore, to learn robust policies from limited demonstrations, we combine two loss terms: i) an RL term trained on an offline dataset $\mathcal{D}^O$, weighted by the advantage scores; and ii) a BC term trained on a smaller set of expert trajectories $\mathcal{D}^E$, providing a stabilizing prior:

$$\mathcal{L}_\pi^{\text{RL}}(\phi) = \frac{1}{H}\sum_{h=1}^{H}\sum_i\mathbb{E}_{\mathcal{D}^O}\left[\omega_i^{h:h+k}(s^h, a_i^{h:h+k})\left\|\epsilon - \pi_\phi(\sqrt{\hat{\alpha}}a_i^{h:h+k} + \sqrt{1-\hat{\alpha}}\epsilon, s^h, t)\right\|\right], \quad (8)$$

$$\mathcal{L}_\pi^{\text{BC}}(\phi) = \frac{1}{H}\sum_{h=1}^{H}\sum_i\mathbb{E}_{\mathcal{D}^E}\left[\left\|\epsilon - \pi_\phi(\sqrt{\hat{\alpha}}a_i^{h:h+k} + \sqrt{1-\hat{\alpha}}\epsilon, s^h, t)\right\|\right], \quad (9)$$

$$\mathcal{L}_\pi(\phi) = \mathcal{L}_\pi^{\text{RL}}(\phi) + \beta\mathcal{L}_\pi^{\text{BC}}(\phi), \quad (10)$$

where $\epsilon \sim \mathcal{N}_{\text{pol}}(0, I)$ denotes Gaussian noise, $t$ is the noising timestep, $\hat{\alpha}_t$ represents the noise schedule parameter in diffusion training, and $\beta > 0$ controls the trade-off between reinforcement learning-driven exploration and expert imitation.

## 4.3 General Algorithms and Practical Implementation

Algorithm 1 offers an overview of the HVD approach, which operates in two phases. The first phase focuses on hierarchical value learning, where both the value network $V_\psi$ and Q-network $Q_\theta$ are updated using the hierarchical value-decomposed learning loss (Equation 6) and TD loss (Equation 5), respectively. The second phase performs policy extraction, where the policy network $\pi_\phi$ is trained to maximize the cumulative returns through advantage weighted regression (Equation 10).

**Model Architecture.** As illustrated in Figure 21, our hierarchical Q-network adopts a unified multi-modal architecture centered around a Transformer-based backbone. The model can optionally process a rich set of sensory modalities as input by processing them into token embeddings, including egocentric RGB images, point cloud data from depth sensors, natural language task instructions, and proprioceptive state. More detailed implementation can be found in Appendix C.3.

## 5 EXPERIMENTS

Our experiments aim to address three core questions. **Q1**: Does HVD consistently surpass imitation learning baselines across diverse policy architectures? (Section 5.2) **Q2**: Does hierarchical value decomposition yield more accurate credit assignment, and does this improve the policy performance? (Section 5.3) **Q3**: Can HVD effectively scale to multi-task settings, leveraging shared value structure to improve overall performance? (Section 5.4)

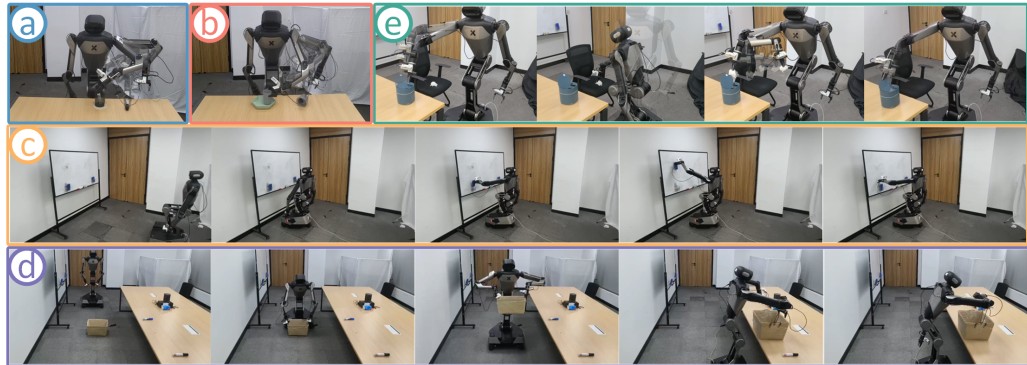

Figure 3: Illustration of evaluated tasks: (a) *Pen Insert*, (b) *Cup Upright*, (c) *Wipe Board*, (d) *Basket Carry*, (e) *Trash Dispose*.

### 5.1 EXPERIMENT SETTINGS

**Robot Platform.** We conduct all experiments on the Galaxea R1, a real-world wheeled humanoid robot with a 21-DoF whole-body system. Task demonstrations are collected using JoyLo, a teleoperation interface developed by (Jiang et al., 2025). Guidelines are provided to constrain operators to generate demonstrations that are easier for the robot to learn. Nevertheless, operator skill levels vary, resulting in a substantial number of suboptimal demonstrations during data collection.

**Task Design.** We design a suite of five representative office tidying tasks (see Figure 3 with details in Appendix B). The tasks require navigation, dexterous manipulation, and bimanual coordination, with durations from 40-second single-arm actions to **over 120-second** multi-step sequences involving coordinated locomotion and dual-arm cooperation. Additionally, we assess task difficulty from temporal complexity, kinematics, control, and coordination complexity as detailed in Appendix B.6. This diversity enables rigorous testing of both precision and long-horizon whole-body control.

**Baselines.** We develop our HVD framework based on three baselines with different input modalities, including the state-of-the-art VLA model $\pi_0$ (Black et al., 2024), the 3D-input model WB-VIMA (Jiang et al., 2025), and the Diffusion Policy (Chi et al., 2023). We evaluate the performance of policies trained using the original methods on expert datasets and compare them with policies trained using HVD on mixed-quality datasets.

**Evaluation Metrics.** To enable fine-grained assessment of policy performance, each task is decomposed into distinct logical stages. We report two primary metrics: *success rate* for task-level evaluation and *normalized stage score* for stage-level analysis. Moreover, we introduce perturbations to the task environment background, initial task region, and robot's initial pose to further challenge robustness and evaluate the model's ability to generalize under diverse and realistic variations. Each policy was evaluated over 50 independent rollouts per task. Reported success rates and stage scores are averaged over these rollouts to ensure statistical consistency and fair comparison across methods.

### 5.2 BENCHMARK RESULTS

We present the main experimental results of our study, evaluating each method under its optimal training regime to assess peak performance. HVD is trained on the full mixed-quality dataset, while imitation learning baselines are trained on the expert-only subset, consistent with their reliance on high-quality demonstrations.

As shown in Table 1, our proposed **HVD** consistently outperforms standard imitation learning across all five tasks, yielding higher average success rates across policies. The gains are especially pronounced in challenging manipulation tasks such as *Wipe Board* and *Basket Carry*, where robustness to imperfect initial states and partial observability is essential. Moreover, Figure 4 shows that HVD's benefits extend beyond task-level success: it achieves higher normalized stage scores on nearly all subtasks. This demonstrates that HVD not only improves final outcomes but also enhances policy reliability throughout the entire execution trajectory.

| Method | Tasks | | | | | Avg SR |
|---|---|---|---|---|---|---|
| (IL/HVD) | Pen Insert | Cup Upright | Wipe Board | Basket Carry | Trash Dispose | |
| $\pi_0$ | 0.64/**0.86** | 0.82/**0.90** | 0.18/**0.32** | 0.26/**0.44** | 0.28/**0.36** | 0.44/**0.58** |
| WB-VIMA | 0.52/**0.78** | 0.58/**0.82** | 0.12/**0.26** | 0.10/**0.10** | 0.20/**0.32** | 0.30/**0.46** |
| DP | 0.54/**0.64** | 0.66/**0.72** | 0.00/**0.00** | 0.00/**0.08** | 0.08/**0.16** | 0.26/**0.32** |

Table 1: Task-level success rate (SR) of IL and HVD across baseline methods on 5 tasks.

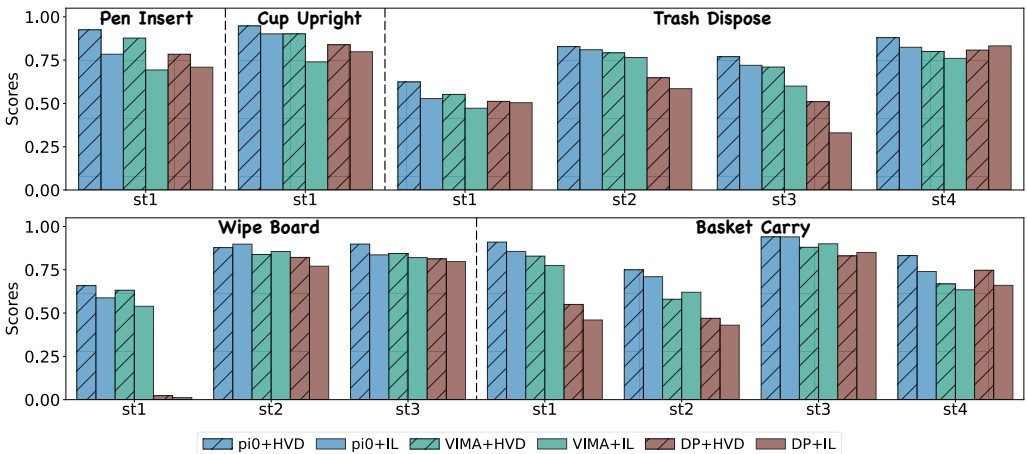

Figure 4: Stage-level scores of IL and HVD across baseline methods across 5 tasks.

## 5.3 VALUE DECOMPOSITION ABLATION

First, we investigate whether the observed performance gain is primarily attributable to the hierarchical value decomposition or merely to the application of offline RL. To this end, we conduct an ablation study comparing our model with the shared Q-value (named **HVD w/o hierarchy**). Both models are trained on the same mixed-quality dataset with the same hyperparameters, ensuring that the only architectural difference is the presence of hierarchical value decomposition. Table 2 shows that removing hierarchical decomposition consistently harms performance across tasks, confirming that the improvements not only stem from the training paradigm, but also from the proposed structure.

| Method | Tasks | | | | | Avg Diff |
|---|---|---|---|---|---|---|
| w/o hierarchy | Pen Insert | Cup Upright | Wipe Board | Basket Carry | Trash Dispose | |
| DP | -0.02 | 0.00 | 0.00 | -0.08 | -0.06 | -0.03 |
| WB-VIMA | -0.02 | 0.00 | -0.12 | -0.08 | -0.12 | -0.07 |
| $\pi_0$ | +0.04 | -0.02 | -0.14 | -0.10 | -0.04 | -0.05 |

Table 2: Ablation study results on hierarchical value decomposition across 5 tasks. The value here indicates the success rate changes when removing the hierarchy.

To further analyze how decomposition impacts credit assignment, we visualize the advantage weights $\omega_i$ of several key frames in the *Basket Carry* task (Figure 5). At the second key frame, where the robot prepares to stand and hold the basket, HVD assigns higher weights to the arm and torso, reflecting the importance of these components. In contrast, HVD w/o Hierarchy produces uniformly high weights across all frames, failing to differentiate subcomponents. These findings indicate that HVD enables more precise credit assignment, which directly contributes to more accurate and reliable action generation. The more visualization examples are shown in Appendix D.1.

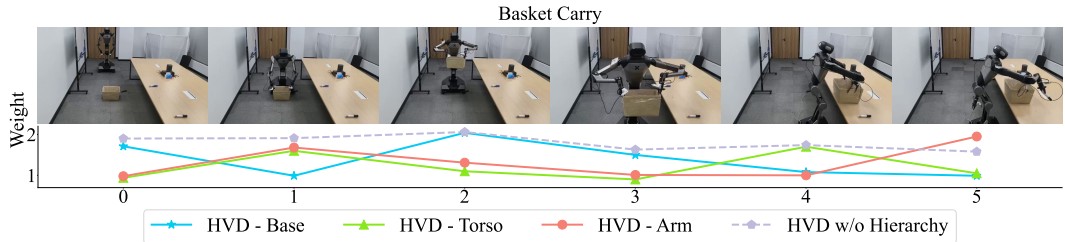

Figure 5: **Credit Assignment Comparison** between HVD and HVD w/o hierarchy.

## 5.4 MULTI-TASK LEARNING

We evaluate HVD's ability to scale to multi-task learning by training a single policy on data from all five tasks and comparing its success rate against specialist (single-task) policies. As shown in Table 3, standard IL suffers from negative transfer in the multi-task setting, leading to degraded performance on most tasks. In contrast, HVD mitigates such interference and even surpasses single-task specialists on several tasks. We observe that these gains primarily stem from *enhanced torso robustness* and *more generalizable grasping behaviors* acquired during multi-task training. Together, these results show that HVD effectively leverages shared knowledge across tasks while maintaining specialization, making it more scalable to diverse multi-task settings.

| Method | Tasks | | | | | Avg SR |
|---|---|---|---|---|---|---|
| $\pi_0$ | Pen Insert | Cup Upright | Wipe Board | Basket Carry | Trash Dispose | |
| expertise IL | 0.64 | 0.82 | 0.18 | 0.26 | 0.28 | 0.44 |
| multi-task IL | 0.50 | 0.60 | 0.18 | 0.24 | 0.30 | 0.36 (-0.08) |
| expertise HVD | 0.86 | 0.90 | 0.32 | 0.44 | 0.36 | 0.58 |
| multi-task HVD | 0.92 | 0.94 | 0.32 | 0.50 | 0.30 | 0.60 (+0.02) |

Table 3: Task-level success rate comparison of multi-task and expertise policies across 5 tasks.

## 6 RELATED WORK

**Whole-body Control Policy Learning.** Whole-body control is a central challenge in robotics due to the high dimensionality of articulated bodies and the lack of inherent self-stabilization (Hirai et al., 1998; Grizzle et al., 2009). Classical model-based planning emphasizes kinematic feasibility, stability, and reactive regulation (Sentis & Khatib, 2006; Dietrich et al., 2012; Burget et al., 2013; Kaelbling & Lozano-Pérez, 2013; Dai et al., 2014), but struggles with adaptability and scalability in unstructured tasks. Learning-based approaches optimize control policies from data (Siekmann et al., 2021; Li et al., 2021; Dao et al., 2022; Radosavovic et al., 2024; Cheng et al., 2024), enabling dynamic behaviors difficult to engineer manually (Xia et al., 2021; Jiang et al., 2024; Fu et al., 2024; Arm et al., 2024). Recent advances include generative policies for capturing multimodal action distributions (Fu et al., 2023; Jiang et al., 2025), VLA models that ground control in language and perception (Xu et al., 2024b; Ding et al., 2025), and hierarchical policy architectures for managing the complexity of humanoid whole-body control (Hansen et al., 2025; Wei et al., 2025; Fu et al., 2025). Despite these advances, most methods still require high-quality demonstrations, limiting scalability to complex real-world tasks.

**Offline RL for Embodied Control.** Offline RL has made significant strides in embodied control tasks, enabling robots to learn complex behaviors from pre-collected datasets without requiring expert demonstrations (Levine et al., 2020; Lin et al., 2025; 2026). Previous works have attempted to learn policies from trajectories generated by human failures or during policy evaluation (Kumar et al., 2021; Mandlekar et al., 2022; Sinha et al., 2022; Bhateja et al., 2023; Luo et al., 2023; Zhou et al., 2023; Zhang et al., 2023b; Ma et al., 2024; Zhang et al., 2024). However, these methods are largely confined to arm-based manipulators, and their effectiveness in high-DoF whole-body control tasks remains unexplored. Recently, there has been an effort to adapt RL algorithms to mobile manipulators (Hu et al., 2023; Pan et al., 2024). However, these approaches often rely on single-modality inputs, limiting their ability to integrate with generalist policies like VLA and world models (Kim et al., 2024; Black et al., 2024; Zhang et al., 2023a; 2024). Furthermore, most of these methods focus on single-task training, raising concerns about their scalability and generalization to multi-task scenarios.

## 7    Conclusions and Limitations

**Conclusions.** In this paper, we present HVD, a framework for learning whole-body robotic control from imperfect, real-world demonstrations. By introducing kinematically aware value decomposition within a multi-modal Transformer architecture, HVD enables stable and scalable policy learning in high-dimensional action spaces using suboptimal offline data. Together with the release of WB-50, a 50-hour dataset of realistic teleoperation and rollout trajectories, we demonstrate that structured offline RL can effectively leverage partial successes and failures to achieve robust, generalizable control. The information about resource cost is listed in Appendix E.

**Limitations.** First, HVD relies on human-annotated rewards, which can be costly. Second, we have not yet explored using failed data for pretraining, which could become a valuable paradigm for improving robustness and scalability in open-ended environments. Future work could also investigate leveraging VLM for automated reward labeling to reduce human effort and enhance scalability.

## Acknowledgments

This work was supported by the National Natural Science Foundation of China under Grants 62495090, 62495093, U23B2059, 62506159, U24A20324, and the Natural Science Foundation of Jiangsu under Grants BK20241199, BK20243039.

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

# A  HARDWARE COMPONENTS

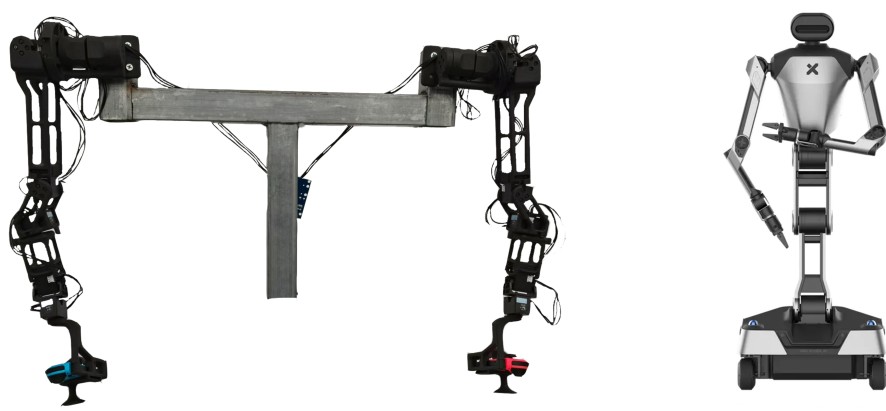

Figure 6: Low-cost JoyLo system and Galaxea R1 robot.

As illustrated in Figure 6, the hardware setup comprises a JoyLo system and a Galaxea R1 robot. The JoyLo system integrates 3D-printable arm links, low-cost Dynamixel actuators, and commodity Joy-Con controllers; its control loop runs at 100 Hz while data are recorded at 10 Hz. Functional buttons on the right Joy-Con are used to start, pause, save, and discard recordings. Logged modalities include RGB and depth images, point clouds, joint states, odometry, and action commands. The Galaxea R1 platform is equipped with a ZED 2i stereo camera, two Intel RealSense D435i cameras, and two Galaxea G1 parallel grippers.

# B  TASK DEFINITION

## B.1  PEN INSERT

**Task Description**   This task requires the robot to grasp a marker pen lying on a table and insert it vertically into a fixed pen holder (diameter 8 cm). The core challenge lies in seamlessly executing the entire sequence from grasping to insertion. For each trial, both the marker pen and pen holder are randomly positioned on the tabletop within the robot's operational workspace and field of view. To further validate generalization, the robot's starting position and torso pose are also randomized within a constrained range.

**Evaluation Rubric**   The task is evaluated as a single, continuous stage focusing on the successful transfer of the pen to the holder.

**Stage 1: Grasp and Insert Pen into Holder**

- `0.0 points`: The robot fails to grasp the marker pen (left of Figure 7).

- `0.5 points`: The robot successfully grasps the marker pen but fails to place it in the holder, for instance, by dropping the pen outside the holder due to an insecure hold (middle of Figure 7).

- `1.0 points`: The robot firmly grasps the marker pen and successfully places it into the holder (right of Figure 7).

**Prompt:**  `pick the pen and put it into the holder`

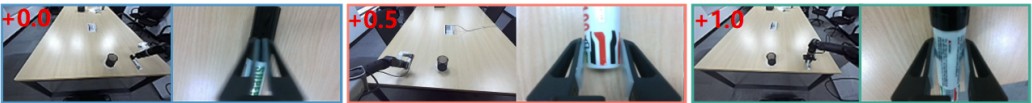

Figure 7: Scoring rubric visualization for *Pen Insert*. Stage 1: Grasp and Insert Pen into Holder

## B.2 CUP UPRIGHT

**Task Description.**  This task requires the robot to grasp a horizontally lying plastic cup (diameter 8 cm) and place it in a stable, upright position onto a target plate. The primary challenges involve dexterous reorientation of the cup during manipulation and ensuring a steady final placement. For each trial, the cup is randomly placed on the tabletop, and the robot's starting pose is randomized within a constrained range to test for policy generalization. The task demands a combination of precise grasping and controlled, stable placement.

**Evaluation Rubric**  The task is evaluated as a single, continuous stage that assesses the entire sequence from grasping to successful upright placement.

**Stage 1: Grasp and Place Cup Upright**

- `0.0 points`: The robot fails to secure the cup with its gripper (left of Figure 8).
- `0.5 points`: The robot grasps the cup but fails to place it upright on the plate, either due to losing its grip or improper reorientation (middle of Figure 8).
- `1.0 points`: The robot firmly grasps the cup and places it steadily in an upright position on the plate (right of Figure 8).

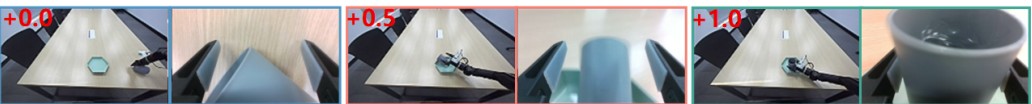

Figure 8: Scoring rubric visualization for *Cup Upright*. Stage 1: Grasp and Place Cup Upright

**Prompt:** `pick the cup and put it onto the coaster`

## B.3 WIPE BOARD

**Task Description.**  This task is composed of three sequential stages. In the first stage, the robot navigates its base to a position in front of the whiteboard and grasps an eraser. The second stage involves wiping the designated markings from the board. In the third and final stage, the robot places the eraser back into its designated slot. To promote policy generalization, each trial is initialized with randomized starting poses for the robot and varied positions for the writing on the whiteboard, both within a predefined area. This task demands precise physical interaction, as successful execution hinges on delicate force control: excessive pressure may cause the whiteboard to tilt, while insufficient force will fail to clean the markings completely.

**Stage 1: Approach Whiteboard and Grasp Eraser**

- `0.0 points`: The robot fails to navigate to a position where any eraser is reachable (left of Figure 9).
- `0.5 points`: The robot successfully navigates to the whiteboard but fails to establish a stable grasp on an eraser (middle of Figure 9).
- `1.0 points`: The robot successfully navigates to the whiteboard and executes a stable grasp on an eraser, suitable for the wiping motion (right of Figure 9).

**Stage 2: Wipe Markings**

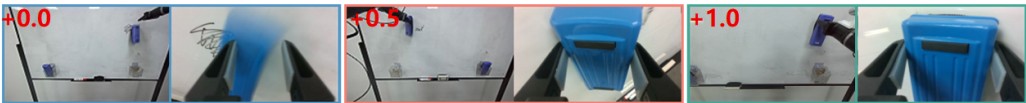

Figure 9: Scoring rubric visualization for *Wipe Board*. Stage 1: Approach Whiteboard and Grasp Eraser

- `0.0 points`: The robot fails to make effective contact with the markings, due to dropping the eraser or significant positioning errors (left of Figure 10).
- `0.5 points`: The robot partially erases the markings, or causes the whiteboard to tilt due to imprecise force control or positioning error (middle of Figure 10).
- `1.0 points`: The robot completely erases the markings while maintaining stable contact with the whiteboard (right of Figure 10).

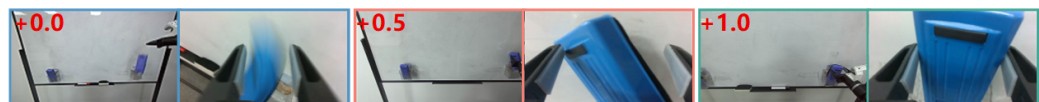

Figure 10: Scoring rubric visualization for *Wipe Board*. Stage 2: Wipe Markings

**Stage 3: Return Eraser**

- `0.0 points`: The robot releases the eraser prematurely before reaching the designated slot (left of Figure 11).
- `0.5 points`: The robot reaches the slot but fails to place the eraser correctly due to positioning inaccuracies, causing it to be dropped (middle of Figure 11).
- `1.0 points`: The robot successfully and stably places the eraser back into its slot (right of Figure 11).

Figure 11: Scoring rubric visualization for *Wipe Board*. Stage 3: Return Eraser

**Prompt:** `move to the whiteboard and clean the whiteboard`

### B.4 BASKET CARRY

**Task description.** This task consists of four distinct stages. First, the robot navigates to a cuboid basket placed on the ground. In the second stage, it bends down and lifts the basket using coordinated movements of both arms. For the third stage, the robot turns while holding the basket and places it onto a nearby table. In the final stage, the robot utilizes its left and right arms respectively to pick up markers from the table and place them into the basket. To promote policy generalization, each trial is initialized with the robot at a randomized starting pose, the basket at a varied position on the ground, and the markers at randomized locations on the table, all within a predefined area. This task demands long-horizon planning and effective bimanual coordination.

**Stage 1: Approach Basket**

- `0.0 points`: The robot navigates to an incorrect position, rendering the basket unreachable for lifting (left of Figure 12).
- `0.5 points`: The robot navigates to a misaligned position, preventing a symmetric, bimanual grasp required for a stable lift (middle of Figure 12).

- `1.0 points`: The robot successfully navigates to a centered position directly in front of the basket, enabling a symmetric, bimanual lift (right of Figure 12).

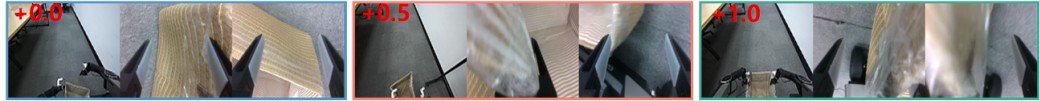

Figure 12: Scoring rubric visualization for *Basket Carry*. Stage 1: Approach Basket

**Stage 2: Lift Basket**

- `0.0 points`: The robot fails to lift the basket off the ground (left of Figure 13).
- `0.5 points`: The robot lifts the basket with both hands but fails to keep it level or properly centered with its body (middle of Figure 13).
- `1.0 points`: The robot successfully lifts the basket, maintaining a level and centered orientation relative to its body (right of Figure 13).

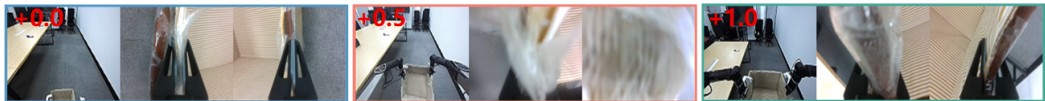

Figure 13: Scoring rubric visualization for *Basket Carry*. Stage 2: Lift Basket

**Stage 3: Place Basket on Table**

- `0.0 points`: The robot fails to place the basket onto the table surface (left of Figure 14).
- `0.5 points`: The basket is placed on the table but is either dropped from a height or left significantly misaligned with the table's edge (middle of Figure 14).
- `1.0 points`: The robot smoothly and squarely places the basket onto the table (right of Figure 14).

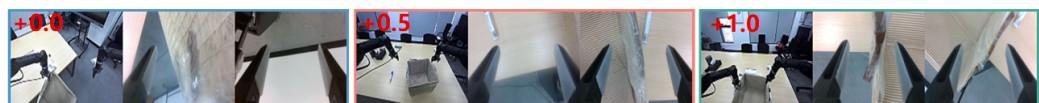

Figure 14: Scoring rubric visualization for *Basket Carry*. Stage 3: Place Basket on Table

**Stage 4: Place Markers in Basket**

- `0.0 points`: The robot fails to place any of the markers into the basket (left of Figure 15).
- `0.5 points`: The robot successfully places the marker from one side into the basket (middle of Figure 15).
- `1.0 points`: The robot successfully places both markers into the basket (right of Figure 15).

**Prompt:** `move to the storage box, pick up the storage box and place it on the table, then put the pen on the table into the box`

## B.5   TRASH DISPOSE

**Task Description**   This task is composed of four sequential stages, designed to evaluate the robot's capability in long-horizon planning and whole-body coordination within a practical cleanup scenario.

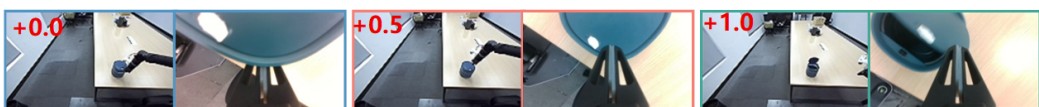

Figure 15: Scoring rubric visualization for *Basket Carry*. Stage 4: Place Markers in Basket

In the first stage, the robot presses the top of a tabletop trash can to open its spring-loaded lid. In the second stage, it turns its body to the right and bends down to grasp a crumpled paper towel placed on a nearby chair, an action requiring substantial whole-body coordination to maintain balance. For the third stage, the robot turns back to the left, moves its arm above the trash can opening, and releases the paper towel. In the final stage, the robot must close the lid and press it down again to lock it in place.

To promote policy generalization, each trial initializes with the robot in a randomized starting pose and the paper towel at a varied position on the chair, both within predefined areas. This task presents a significant challenge due to its extended, multi-stage nature. It demands seamless transitions between pressing, grasping, placing, and locking sub-tasks, all while executing complex, coordinated movements.

**Stage 1: Open Trash Can Lid**

- 0.0 points: The robot fails to open the trash can lid (left of Figure 16).
- 0.5 points: The robot opens the lid but does not succeed on the first attempt (middle of Figure 16).
- 1.0 points: The robot successfully opens the lid on the first attempt (right of Figure 16).

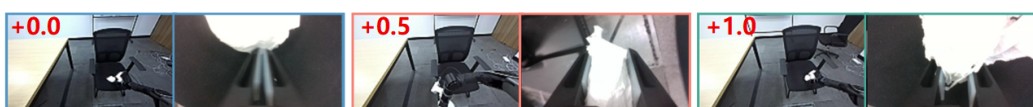

Figure 16: Scoring rubric visualization for *Trash Dispose*. Stage 1: Open Trash Can Lid

**Stage 2: Grasp Paper Towel**

- 0.0 points: The robot fails to grasp the paper towel from the chair (left of Figure 17).
- 0.5 points: The robot grasps the paper towel but not on the first attempt (middle of Figure 17).
- 1.0 points: The robot successfully grasps the paper towel on the first attempt (right of Figure 17).

Figure 17: Scoring rubric visualization for *Trash Dispose*. Stage 2: Grasp Paper Towel

**Stage 3: Dispose of Paper Towel**

- 0.0 points: The robot fails to dispose of the paper towel into the trash can (left of Figure 18).
- 0.5 points: The paper towel lands on the rim or gets stuck at the edge of the trash can during disposal (middle of Figure 18).
- 1.0 points: The robot successfully disposes of the paper towel into the trash can (right of Figure 18).

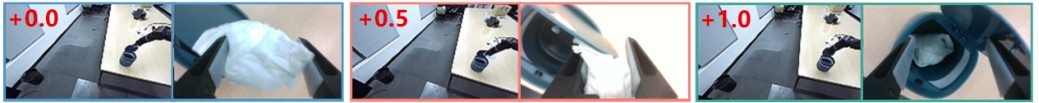

Figure 18: Scoring rubric visualization for *Trash Dispose*. Stage 3: Dispose of Paper Towel

**Stage 4: Close and Lock Trash Can Lid**

- `0.0 points`: The robot fails to close the lid of the trash can (left of Figure 19).
- `0.5 points`: The robot pushes the lid down but fails to press it again to lock it, causing the lid to remain unlatched (middle of Figure 19).
- `1.0 points`: The robot successfully closes the lid and presses it to ensure it is securely locked (right of Figure 19).

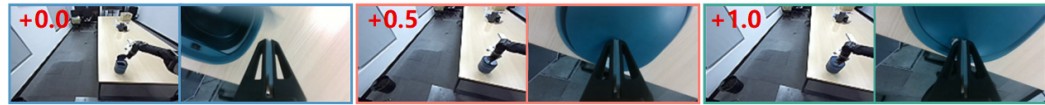

Figure 19: Scoring rubric visualization for *Trash Dispose*. Stage 4: Close and Lock Trash Can Lid

**Prompt:** `Open the trash bin, turn around, pick up the trash on the chair and put it into the bin, then close the bin.`

### B.6 TASK COMPLEXITY METRIC

To provide a quantitative and objective measure of difficulty for our task suite, we compute a composite complexity score for each task from the collected expert demonstration data ($\mathcal{D}_{\text{expert}}$). This score is derived from four distinct metrics, each capturing a different aspect of task complexity. Let a single expert trajectory be a sequence of 21-DoF joint states $\{q_t\}_{t=1}^T$. The metrics are calculated for each trajectory and then averaged across all demonstrations for a given task.

**Temporal Complexity ($C_{\text{time}}$)** This metric captures the temporal length of the task and is calculated as the average duration in seconds over all expert demonstrations.

$$C_{\text{time}} = \frac{1}{N} \sum_{i=1}^{N} T_i \cdot \Delta t, \tag{11}$$

where $N$ is the number of expert trajectories and $\Delta t$ is the time step duration.

**Kinematic Complexity ($C_{\text{kinematic}}$)** This metric quantifies the total magnitude of motion required, which is distinct from temporal duration. It is calculated as the average sum of the $\ell_1$ norm of joint displacements between consecutive timesteps, capturing the spatial extent of the behavior.

$$C_{\text{kinematic}} = \frac{1}{N} \sum_{i=1}^{N} \sum_{t=1}^{T_i-1} \|q_{t+1}^{(i)} - q_t^{(i)}\|_1, \tag{12}$$

where $q_t^{(i)}$ is the joint state vector for the $i$-th trajectory at time $t$.

**Control Complexity ($C_{\text{control}}$)** This metric serves as a proxy for control difficulty by measuring the lack of smoothness in the motion. We approximate the average total jerk using the third-order finite difference of the joint positions.

$$C_{\text{control}} = \frac{1}{N} \sum_{i=1}^{N} \sum_{t=2}^{T_i-2} \|(q_{t+2}^{(i)} - 3q_{t+1}^{(i)} + 3q_t^{(i)} - q_{t-1}^{(i)})\|_2, \tag{13}$$

Higher values indicate more frequent changes in acceleration, suggesting a higher demand on the controller.

**Coordination Complexity ($C_{\textbf{coord}}$)** This metric estimates the number of joints actively involved in the task. For each trajectory, we compute the variance $\sigma_j^2$ for each of the 21 joints over time. The effective dimensionality is the average number of joints whose variance exceeds a small threshold $\epsilon$ (e.g., $10^{-4}$).

$$C_{\text{coord}} = \frac{1}{N} \sum_{i=1}^{N} \sum_{j=1}^{21} \mathbb{I}(\sigma_j^2 > \epsilon). \tag{14}$$

where $\mathbb{I}(\cdot)$ is the indicator function.

**Complexity Score** To synthesize these individual metrics into a single, comparable score for each task, we first perform a min-max normalization on each of the four metrics across the entire task suite. This procedure scales the values of each metric to the range of [0, 1], ensuring that each component contributes equally to the final score regardless of its original units or scale. The final composite complexity score, $C_{\text{final}}$, is then computed as the unweighted average of these four normalized scores, providing a holistic and unified measure of task difficulty. The results of this analysis are presented in Table 4.

Table 4: Quantitative analysis of task complexity. All presented metric scores ($C_k$) have been min-max normalized to the range [0, 1] for direct comparison. The final score is the unweighted average of these normalized values, confirming our task suite spans a graduated range of difficulty.

| Task | Temporal ($C_{\text{time}}$) | Kinematic ($C_{\text{kinematic}}$) | Control ($C_{\text{control}}$) | Coordination ($C_{\text{coord}}$) | Final Score ($C_{\text{final}}$) |
|---|---|---|---|---|---|
| Pen insert | 0.00 | 0.00 | 0.05 | 0.01 | **0.02** |
| Cup upright | 0.06 | 0.51 | 0.00 | 0.00 | **0.14** |
| Wipe board | 0.63 | 0.73 | 1.00 | 0.06 | **0.61** |
| Basket carry | 0.47 | 1.00 | 0.86 | 0.78 | **0.78** |
| Trash dispose | 1.00 | 0.92 | 0.36 | 1.00 | **0.82** |

### B.7 DATASET DESCRIPTION

The WB-50 dataset is a reward-labeled whole-body manipulation dataset spanning over 50 hours of diverse robot experience. WB-50 contains three data sources to reflect realistic data distributions: (i) expert demonstrations ($43.7\%$), (ii) imperfect teleoperation ($14.6\%$), and (iii) policy rollouts ($41.7\%$) — the latter two comprising the majority, mirroring the scarcity of perfect supervision in practice. The proportion of the five task data frames to the total number of frames in the dataset is shown in the Figure 20. It also shows the successful and failed trajectories generated during the expert data collection process of each task, and the number of frames of successful and failed trajectories generated by imitation learning strategy reasoning is also shown. Moreover, we assign rewards of 0, 0.5, and 1 at the end of each subtask according to the degree of task completion, and apply a step penalty of -0.001 for all other steps.

## C IMPLEMENTATION DETAILS

### C.1 ALGORITHM PIPELINE

We present the pseudocode of our method as in Algorithm 1.

### C.2 IMITATION LEARNING BASELINES

**WB-VIMA.** Our implementation of WB-VIMA is based on the official policy codebase (Jiang et al., 2025) and applies minor adjustments to the model and training hyperparameters. The detailed parameters are summarized in Table 5.

**Diffusion Policy.** We build our diffusion policy on the official GalaxeaDP codebase (Team, 2025), which demonstrates strong compatibility with the Galaxea R1 robot, also developed by Galaxea. As

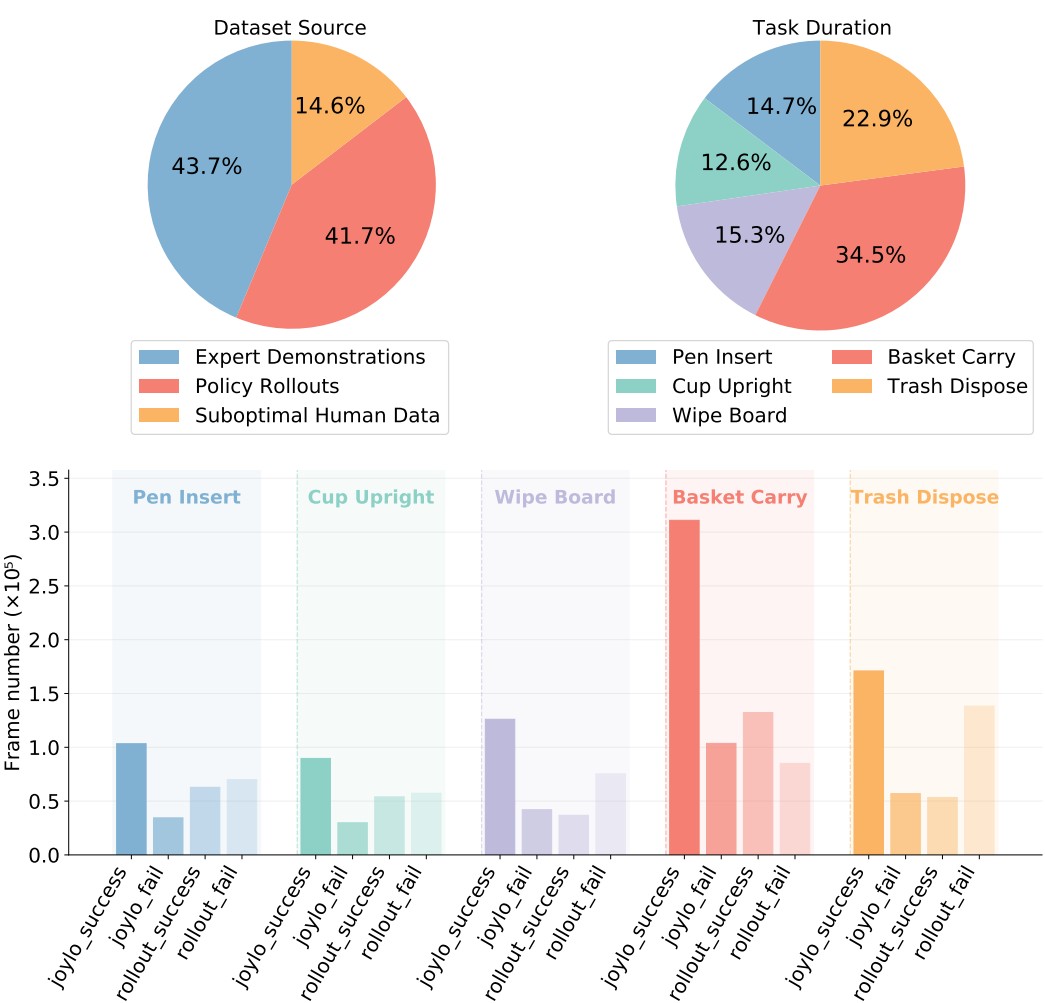

Figure 20: Data distribution of **WB-50**. Distribution of WB-50. The plot shows the proportional composition of the dataset by source, including expert demonstrations, suboptimal expert trajectories and rollout data, together with counts of frames of successful and failed episodes collected during data acquisition.

---

**Algorithm 1** Hierarchical Value-Decomposed Offline Reinforcement Learning (HVD)

---

1: **Input:** Offline dataset $\mathcal{D}$, action hierarchy: $\mathcal{A} = \mathcal{A}_{\text{base}} \times \mathcal{A}_{\text{torso}} \times \mathcal{A}_{\text{arm}}$
2: Initialize value network $V_\psi$, Q-network $Q_\theta$, policy network $\pi_\phi$
   {**Phase 1: Hierarchical Value Learning**}
3: **for** each gradient step **do**
4:     (Update value network)
5:     $\psi \leftarrow \psi - \lambda_V \nabla_\psi \mathcal{L}_V(\psi)$ by Equation equation 6
6:     (Update Q-network)
7:     $\theta \leftarrow \theta - \lambda_Q \nabla_\theta \mathcal{L}_Q(\theta)$ by Equation equation 5
8: **end for**
   {**Phase 2: Policy Extraction**}
9: **for** each gradient step **do**
10:     (Update policy network)
11:     $\phi \leftarrow \phi - \lambda_\pi \nabla_\phi \mathcal{L}_\pi(\phi)$ by Equation equation 10
12: **end for**
13: **Output:** Trained policy $\pi_\phi$

---

Table 5: Hyperparameters of WB-VIMA model.

| Hyperparameter | Value | Hyperparameter | Value | Hyperparameter | Value |
|---|---|---|---|---|---|
| PointNet | | Prop. MLP | | Transformer | |
| $N_{pcd}$ | 4096 | Input Dim | 21 | Embed Size | 512 |
| Hidden Dim | 256 | Hidden Dim | 256 | Num Layers | 8 |
| Hidden Depth | 2 | Hidden Depth | 3 | Num Heads | 8 |
| Output Dim | 256 | Output Dim | 256 | Drop Rate | 0.1 |
| Activation | GELU | Activation | ReLU | Activation | GEGLU |

Table 6: Hyperparameters of WB-VIMA training process.

| Hyperparameter | Value |
|---|---|
| Learning Rate | $1 \times 10^{-4}$ |
| Weight Decay | 0.1 |
| Learning Rate Warm Up Steps | 1000 |
| Learning Rate Cosine Decay Steps | 300,000 |
| Minimal Learning Rate | $5 \times 10^{-6}$ |

the original implementation supports only 14-DoF dual-arm tasks, we extend it by incorporating additional control for the torso and mobile base to enable 21-DoF whole-body control tasks. The hyperparameters and model architectures used in our experiments are summarized in Table 7.

| Hyperparameter | Value |
|---|---|
| Batch Size | 32 |
| Chunk Size | 20 |
| History Size | 2 |
| Learning Rate | 1e-4 |
| LR Scheduler | cosine |
| Optimizer | AdamW |
| AdamW Betas | [0.9, 0.95] |
| Weight Decay | 1e-4 |
| Max Training Steps | 100,000 |
| Image Type | RGB |
| Egocentric Perception Type | Joint |
| Observation Encoder | ResNet-18 (He et al., 2016) |
| Diffusion Model | DDPM (Ho et al., 2020) |
| Diffusion Steps | 20 |
| Diffusion Network | U-Net (Ronneberger et al., 2015) |
| U-Net Structure | [256, 1024, 4096] |

Table 7: Hyperparameters of Diffusion Policy.

$\pi_0$. We adopt the official implementation of $\pi_0$ (Black et al., 2024) as our codebase. Key hyperparameters are listed in Table 8.

## C.3 HVD IMPLEMENTATION DETAILS

**Model Architecture.** HVD is implemented using a transformer backbone, which naturally accommodates multiple input modalities. Specifically, the model processes observations through specialized encoders for each modality:

- **RGB Input:** Three egocentric RGB views (front, left, and right) are independently processed using SigLIP (Zhai et al., 2023), producing sequences of visual tokens that capture spatial context and object semantics.

| Hyperparameter | Value |
|---|---|
| Batch Size | 32 |
| Chunk Size | 20 |
| Learning Rate | 1e-4 |
| LR Scheduler | cosine |
| Optimizer | AdamW |
| AdamW Betas | [0.9, 0.95] |
| AdamW Epsilon | 1e-8 |
| Weight Decay | 1e-10 |
| Max Training Steps | 50,000 |
| Fine-tune Method | LoRA (Hu et al., 2022) |

Table 8: Hyperparameters of $\pi_0$.

- **Point Cloud Input:** Depth-derived point clouds are encoded with PointNet (Qi et al., 2017) modules, enabling robust perception of 3D geometry and scene layout, which is particularly beneficial for navigation and object manipulation.

- **Task Instruction:** Natural language commands (e.g., "clean the whiteboard") are tokenized and embedded to provide high-level goal guidance.

- **Proprioception:** Joint angles, velocities, and end-effector poses are concatenated and normalized to form a compact state vector representing the robot's internal configuration.

The tokens produced by all modalities are concatenated along the sequence dimension to form a unified representation. Attention masks regulate cross-modal interactions, after which the integrated token sequence is processed by a pretrained PaliGemma model (Beyer et al., 2024). The resulting representations are then passed through an MLP-based value decoder to estimate Q-values for the different hierarchical components. The overall model architecture is depicted in Figure 21.

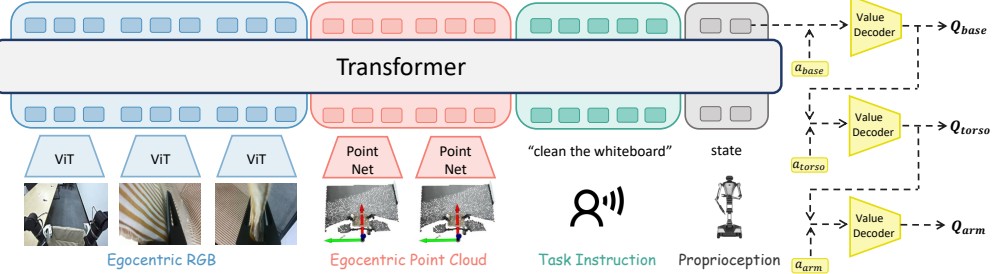

Figure 21: Overall Model Architecture of Hierarchical Q-Network.

**Hyperparameters.** In our implementation, the parameter $\alpha$ in equation 6 controls the relative weighting between the TD loss and the BC loss, effectively balancing value estimation and policy imitation, while $\beta$ in equation 10 is used in the exponential weighting of advantages when computing hierarchical action weights, modulating the sensitivity to high-advantage actions. Both $\alpha$ and $\beta$ are set to their default values of $1.0$. Importantly, we did not perform any hyperparameter tuning, yet our method already achieves strong performance, highlighting the effectiveness and robustness of the proposed HVD approach. Configurations and hyperparameter settings are listed in Table 9.

# D    ADDITIONAL EXPERIMENTS

## D.1    QUALITATIVE VISUALIZATION OF HIERARCHICAL CREDIT ASSIGNMENT

The remaining weight visualization results are presented in Figure 22. We observe that the HVD w/o Hierarchy weights exhibit a trend similar to the HVD arm weights across all tasks. This indicates that while HVD w/o Hierarchy is able to capture key frames in which the arms are about to move, it

| Hyperparameter | Value |
|---|---|
| Value Network | MLP[$128 \times 256 \times 64$] |
| Max Training Steps | 30,000 |
| BC Loss Weight $\alpha$ | 1 |
| Exponential Weight $\beta$ | 1 |
| Image Encoder | SigLIP (Zhai et al., 2023) |
| Point Cloud Encoder | PointNet (Qi et al., 2017) |
| Transformer Model | PaliGemma (Beyer et al., 2024) |
| Width | 256 |
| Depth | 4 |
| MLP Dim | 1024 |
| Number of Heads | 4 |
| Number of KV Heads | 1 |

Table 9: Hyperparameters of HVD.

fails to effectively recognize the contributions of the base and torso. These results further support our conclusion that HVD provides more accurate credit assignment.

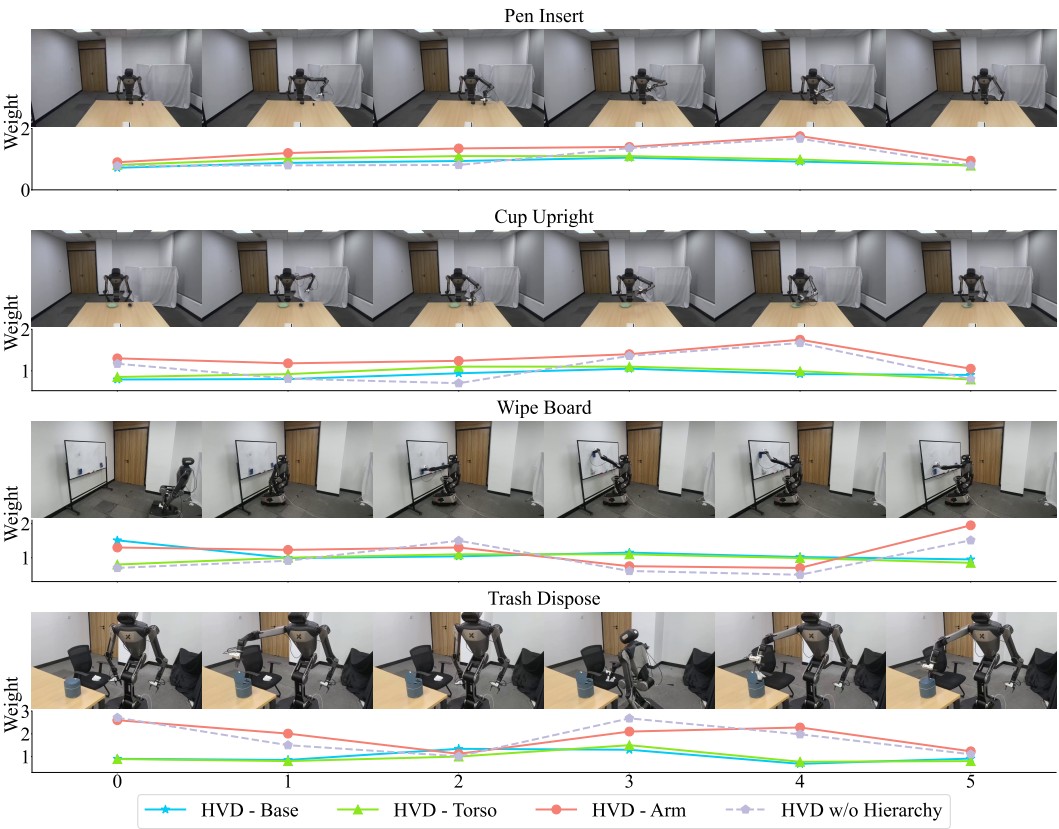

Figure 22: **Remaining Credit Assignment Comparison** between HVD and HVD w/o hierarchy.

## D.2 QUANTITATIVE ANALYSIS OF HIERARCHICAL CREDIT ASSIGNMENT

To validate the interpretability of our hierarchical framework, we analyzed the component weights during policy training across different task stages. The quantitative results in Table 10 reveal distinct stage-aware credit assignment patterns that align with functional requirements. These results

demonstrate that our hierarchical framework successfully differentiates functional requirements across task stages and allocates appropriate credit to different robot components.

Table 10: Hierarchical component weights across task stages.

| Task | Stage | Component Weight | | |
|------|-------|------|------|------|
| | | Base | Torso | Arm |
| Wipe Board | Stage 1: Approach and Grasp Eraser | 0.962 | 0.884 | 1.321 |
| | Stage 2: Wipe Markings | 0.946 | 0.890 | 1.054 |
| | Stage 3: Return Eraser | 0.857 | 0.826 | 1.149 |
| Basket Carry | Stage 1: Approach and Grasp Basket | 0.980 | 0.999 | 1.405 |
| | Stage 2: Lift Basket | 0.953 | 0.909 | 1.254 |
| | Stage 3: Place Basket on Table | 0.837 | 0.807 | 1.063 |
| | Stage 4: Place Markers in Basket | 0.972 | 0.980 | 1.243 |
| Trash Dispose | Stage 1: Open Trash Can Lid | 0.933 | 0.918 | 1.382 |
| | Stage 2: Grasp Paper Towel | 0.930 | 0.960 | 1.273 |
| | Stage 3: Dispose of Paper Towel | 0.944 | 0.902 | 1.356 |
| | Stage 4: Close and Lock Trash Can Lid | 0.804 | 0.813 | 1.323 |

### D.3 ANALYSIS OF TASK COMPLEXITY AND POLICY PERFORMANCE

By correlating the difficulty metrics presented in Table 4 with the actual success rates of learned policies in Table 1, we can identify several key bottlenecks in current whole-body control approaches and point out the primary sources of performance gains introduced by HVD. Specifically, our analysis reveals that high control complexity and high kinematic coordination demands are the primary failure modes for standard IL, which HVD successfully mitigates.

**Overcoming the Smoothness Bottleneck in High-Control Complexity Tasks.** $C_{\text{control}}$ serves as the most correlated predictor of failure for baseline methods. For instance, the *Wipe Board* task exhibits the maximum Control Complexity ($C_{\text{control}} = 1.00$), indicating a requirement for frequent acceleration changes and high jerk. Standard IL baselines collapse on this task. In contrast, HVD significantly improves $\pi_0$ performance with a $+77\%$ relative improvement, demonstrating that HVD effectively models the non-smooth, high-frequency dynamics often lost in standard IL training.

**Robustness in High-Dimensional Coordination.** IL baselines often fail in the tasks characterized by high $C_{\text{kinematic}}$ and $C_{\text{coord}}$ due to the difficulty of coordinating high-dimensional joints over large spatial displacements. The *Basket Carry* task, which possesses the highest Kinematic Complexity ($C_{\text{kinematic}} = 1.00$) and very high Coordination Complexity ($C_{\text{coord}} = 0.78$), illustrates this barrier. While the WB-VIMA and DP baselines struggle significantly (with DP failing almost completely), HVD provides its most robust improvement here, lifting the $\pi_0$ success rate from $0.26$ to $0.44$. This indicates that HVD acts as a superior regularizer in high-variance regimes, maintaining structural integrity over wide state spaces where standard IL fails to generalize.

### D.4 SIMULATOR EXPERIMENTS

We introduce a simulation experiment on the BEHAVIOR-1K platform to empirically validate our approach, directly addressing the reviewer's query (**W5**) for a concrete simulated environment.

### D.4.1 SIMULATOR PLATFORM

We conduct our empirical evaluation using the BEHAVIOR-1K simulation environment (Li et al., 2024). This platform is a high-fidelity, standardized benchmark focused on humanoid robotics tasks grounded in real-world human needs. The simulated robot model features a base, torso, and arm structure that necessitates complex whole-body coordination, aligning directly with our methodological requirements for addressing the high-Dimensionality of the action space. Figure 23 provides an overview of the simulator setup.

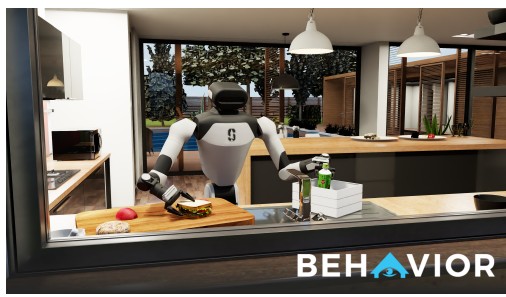 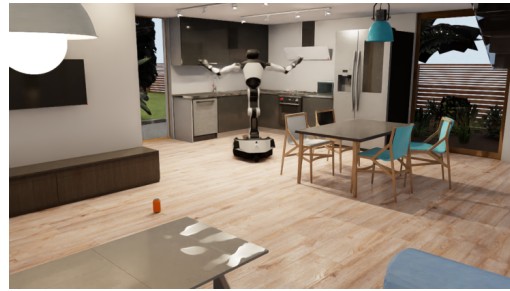

Figure 23: Overview of the simulator in the BEHAVIOR-1K.

**Task Design and Decomposition.** We selected the `Picking Up Trash` task from the official BEHAVIOR-1K benchmark as our test environment. This task highly demands robust whole-body control capabilities. Given the computational intensity of long-horizon rollouts in this complex simulator, we constrained each episode to 10,000 control steps. While this limitation prevents full task completion, it is sufficient to cover the most critical phases. To facilitate focused evaluation and learning, we decomposed the task into two distinct, sequential stages with clear success criteria:

- **Approaching**: Successful if the robot positions itself directly in front of the trash bin, such that the bin is visible in at least one camera view.
- **Grasping**: Successful if the robot lifts the trash bin off the ground using either arm.

For reward labeling, successful trajectories (meeting stage criteria) receive a terminal reward of +1.0, provided at the end of the trajectory. Additionally, every intermediate frame incurs a small step penalty of -0.001 to encourage efficient execution. The overall task pipeline, illustrating this two-stage structure, is presented in Figure 24.

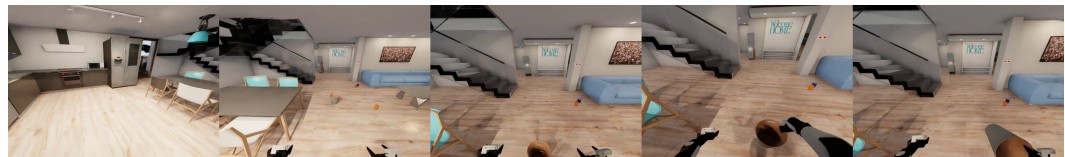

Figure 24: Task pipeline for the Picking Up Trash task in BEHAVIOR-1K, illustrating the approaching and grasping stages evaluated in our experiments.

**Dataset and Evaluation.** Our experimental analysis utilized a training dataset of 300 trajectories: 200 expert demonstrations from the official BEHAVIOR-1K benchmark, supplemented by 100 policy rollouts collected using the pre-trained $\pi_0$-IL baseline. During evaluation, we tested each policy on **100 trials** across 20 unseen scenarios to ensure robustness and generalizability. Table 12 summarizes the performance of the baseline Imitation Learning ($\pi_0$-IL) policy and our Hierarchical Value Decomposition ($\pi_0$-HVD) method across both evaluation stages. HVD consistently demonstrates a significant performance improvement over the baseline:

**Implementation Details.** The HVD implementation was built directly on top of the official $\pi_0$ codebase provided by the BEHAVIOR-1K benchmark, ensuring a fair and consistent comparison with the $\pi_0$, IL baseline. All implementation details, including the hierarchical architecture, network configurations, and hyperparameter settings, remain the same as those introduced in Section C.3, with only minor adjustments as summarized in Table 11.

**Task Results.** Table 12 summarizes the evaluation outcomes, showing that HVD substantially outperforms the IL baseline. In Stage 1, HVD achieves a **77.0%** success rate than IL's **45.0%**. The advantage becomes even more pronounced in the more challenging grasping stage, where HVD nearly doubles the success rate, reaching **22.0%** compared to IL's **5.0%**. These results highlight HVD's effectiveness in handling complex, high-dimensional whole-body coordination tasks.

| Hyperparameter | Value |
|---|---|
| Max Value Training Steps | 10,000 |
| Max Policy Training Steps | 80,000 |
| Hierarchical Order | [[0-2], [3-6], [7-22]] |

Table 11: Adjusted hyperparameters of HVD in BEHAVIOR-1K.

| Method | Stage 1 (SR) | Stage 2 (SR) |
|---|---|---|
| $\pi_0$-IL | 45.0% | 5.0% |
| $\pi_0$-HVD | 77.0% | 22.0% |

Table 12: Results for `Picking Up Trash` task in BEHAVIOR-1K.

## D.5 ABLATION STUDY ON DATA DISTRIBUTION

To further empirically validate the data distribution robustness of HVD, we conduct an additional ablation on the Pen Insert task under varying expert ratios. During training, we keep the total number of demonstrations fixed at 100 with expert ratios of 20%, 50%, and 80%. During training, $\pi_0$+IL uses only the expert demonstrations, while $\pi_0$+HVD uses all available demonstrations, including both expert and suboptimal data. The results are shown in Table 13, where HVD consistently outperforms IL, confirming that HVD maintains its effectiveness across a wide spectrum of data compositions. Due to time constraints, each policy is evaluated over 20 trials.

Table 13: Success rates on the Pen Insert task under varying expert-to-imitation data ratios.

| Method | 20 exp + 80 imp | 50 exp + 50 imp | 80 exp + 20 imp |
|---|---|---|---|
| $\pi_0$+IL | 0.15 | 0.35 | 0.50 |
| $\pi_0$+HVD | **0.55** | **0.60** | **0.75** |

## D.6 COMPARISON WITH OFFLINE RL BASELINES

To further benchmark HVD against other offline RL methods, we implement QIPO (Zhang et al.) using the same base policy $\pi_0$ and train in a multi-task learning setting. For a fair comparison, QIPO and HVD share identical network architectures, optimizer configurations, and training hyperparameters. The only differences lie in: (i) the policy update mechanism, and (ii) the use of hierarchical value decomposition. Specifically, QIPO initializes its policy and Q-networks using BC and TD losses, respectively, followed by iterative policy improvement steps. We adopt the hyperparameter recommendations from the original QIPO paper as our starting point and perform minimal tuning to ensure stable convergence. The final configuration uses $\beta = 1$, $M = 16$ (number of sampled actions), and $K_{\text{renew}} = 10$ (policy renewal interval).

As shown in Table 14, HVD consistently outperforms QIPO across all five tasks in terms of success rate (SR). The improvement is most significant in long-horizon, whole-body control tasks, such as *Basket Carry* and *Trash Dispose*, where accurate credit assignment over extended action sequences is critical. We attribute this advantage to HVD's hierarchical value decomposition, which enables more precise reward propagation and subgoal-aware policy learning.

## D.7 ABLATION STUDY ON DECOMPOSITION ORDER

To validate the impact of different decomposition orders and validate our design choice, we conduct an ablation experiment. Specifically, we compare two alternative sequencing strategies: (i) Arm–Torso–Base and (ii) Base–Arm–Torso, using the same base policy $\pi_0$ across all tasks within a multi-task learning setup. Due to time constraints, the baseline policy is evaluated over 20 trials per task. The results are summarized in the Table 15, clearly demonstrating that our proposed decomposition order outperforms the alternatives, highlighting the importance of the hierarchical structure in our approach.

| Method | Tasks | | | | | Avg SR |
|---|---|---|---|---|---|---|
| $\pi_0$ MultiTask | Pen Insert | Cup Upright | Wipe Board | Basket Carry | Trash Dispose | |
| QIPO | 0.90 | 0.90 | 0.30 | 0.20 | 0.25 | 0.51 |
| HVD | **0.92** | **0.94** | **0.32** | **0.50** | **0.30** | **0.60** |

Table 14: Task-level success rate of HVD and QIPO across 5 tasks.

Table 15: Success rates (SR) for different decomposition orders across tasks.

| Method | Pen Insert | Cup Upright | Wipe Board | Basket Carry | Trash Dispose | Avg SR |
|---|---|---|---|---|---|---|
| Arm–Torso–Base | 0.55 | 0.60 | 0.15 | 0.20 | 0.10 | 0.32 |
| Torso–Arm–Base | 0.80 | 0.80 | 0.20 | 0.35 | 0.20 | 0.47 |
| Base–Torso–Arm (ours) | **0.92** | **0.94** | **0.32** | **0.50** | **0.33** | **0.60** |

## D.8 Comparison with Residual Value Decomposition

To further validate the residual approach versus our proposed independent decomposition, we add an additional baseline across all tasks within a multi-task learning setup, which learns the residual Q-function $Q_{residual}$ using the formulation defined above. Due to time constraints, the baseline policy is evaluated over 20 trials per task. The results are as shown in Table 16, showing that our independent decomposition outperforms the residual variant baseline, achieving a higher average success rate.

Table 16: Comparison between residual and independent decomposition variants of HVD.

| Method | Pen Insert | Cup Upright | Wipe Board | Basket Carry | Trash Dispose | Avg SR |
|---|---|---|---|---|---|---|
| HVD (residual) | 0.75 | 0.80 | 0.25 | **0.50** | **0.35** | 0.53 |
| HVD (ours) | **0.92** | **0.94** | **0.32** | **0.50** | 0.33 | **0.60** |

## E  Resource Cost

### E.1  Reward Labeling Cost

Reward signals are manually annotated according to the task definitions provided in Appendix B. For each demonstration, operators carefully review the recordings from multiple camera views (head, left wrist, and right wrist) and mark the key frames where the robot's actions satisfy predefined scoring criteria. This process is highly labor-intensive: labeling the entire **WB-50** dataset required two well-trained operators working for nearly 30 hours. The labeled data are subsequently used to train HVD, yielding improved policy performance.

### E.2  Computational Cost

All trainings are conducted on RTX 4090 GPU platform. During the value learning stage, Hierarchical networks are trained on 4×RTX 4090 GPUs for approximately 2 hours. For policy learning, LoRA fine-tuning of $\pi_0$ requires around 40 hours on 4×RTX 4090 GPUs, while training the DP takes 1×RTX 4090 GPU for 30 hours. WB-VIMA policies are trained on 2×RTX 4090 GPUs for around 24 hours. For deploying policies, 1×RTX 4090 GPU is required to load model parameters and run action inference.

## F  The Use of Large Language Models (LLMs)

Large language models (LLMs) are used in the preparation of this manuscript for sentence-level editing, including improving grammar, clarity, and readability.

