# OpenReview forum: "Hierarchical Value-Decomposed Offline Reinforcement Learning for Whole-Body Control"
_ICLR.cc/2026/Conference — ICLR 2026 Poster_

### Official Review · Reviewer_xcm5 · 2025-10-15

**Soundness:** 2
**Presentation:** 2
**Contribution:** 2
**Rating:** 4
**Confidence:** 4

**Summary:**

This work presents HVD (Hierarchical Value-Decomposed Offline RL), a framework that learns high-DoF whole-body robot control from imperfect, reward-labeled data. By decomposing value functions along the robot’s kinematic hierarchy and temporal structure, HVD enables precise credit assignment in long-horizon tasks. Built on a Transformer architecture, it supports multi-modal, multi-task learning. The authors also release WB-50, a 50-hour teleoperation and policy rollout dataset capturing natural imperfections. Experiments show HVD significantly outperforms baselines, demonstrating that effective whole-body control can emerge from structured learning on imperfect data.

**Strengths:**

1. The paper addresses an interesting and challenging problem of humanoid whole-body control in high-dimensional settings. It proposes a hierarchical decomposition of the Q-function to effectively handle the high degrees of freedom.

2. The empirical validation includes real-world experiments across multiple tasks, demonstrating the effectiveness of the proposed HVD approach. In addition to the method, the paper also introduces a new dataset used for training.

3. The experimental details are clearly presented and appear to be sufficient for reproducibility.

**Weaknesses:**

1. **Figure clarity**: The pipeline figure (Figure 2) lacks captions, making it difficult to follow and interpret. Clear labeling of modules would improve readability.

2. **Theoretical analysis**: The analysis in Proposition 3.1 is limited to the tabular case and cannot be directly applied to humanoid whole-body control, which involves continuous, partially observed states (e.g., image inputs). The justification should be revised, possibly using covering number arguments for general function approximation. The overall intuition—that higher observation dimensionality increases the need for expert data—is valid. However, the proposed hierarchical method decomposes the action space, not the state space, which may not align with the theoretical bound based solely on state-space cardinality.

3. **Experimental fairness**: The experiments compare the proposed offline RL method (which uses reward information) with pure imitation learning, which does not. This comparison is potentially unfair. The authors should include more offline RL baselines based on generative policies [2, 3] that also utilize rewards for a fair evaluation.

4. **Related work**: Several relevant studies on humanoid whole-body control that also exploit hierarchical structures (e.g., [1]) are missing from the discussion and should be cited for completeness.


[1] Hansen, N., Jyothir, S. V., Sobal, V., LeCun, Y., Wang, X., & Su, H. Hierarchical World Models as Visual Whole-Body Humanoid Controllers. In The Thirteenth International Conference on Learning Representations.
[2] Lu, C., Chen, H., Chen, J., Su, H., Li, C., & Zhu, J. (2023, July). Contrastive energy prediction for exact energy-guided diffusion sampling in offline reinforcement learning. In International Conference on Machine Learning (pp. 22825-22855). PMLR.
[3] Zhang, S., Zhang, W., & Gu, Q. Energy-Weighted Flow Matching for Offline Reinforcement Learning. In The Thirteenth International Conference on Learning Representations.

**Questions:**

1. Could you revise Figure 2 and refine the theoretical analysis section to strengthen the justifications and improve clarity?

2. Could you include additional comparisons with offline RL methods that also utilize reward signals for a fairer evaluation?

---

> ### Author Response · Authors · 2025-11-21
>
> We sincerely appreciate the reviewer’s constructive feedback and would like to offer further clarification in response.
>
> >**W1, Q1**: The pipeline figure (Figure 2) lacks captions, making it difficult to follow and interpret. Clear labeling of modules would improve readability.
>
> We apologize for the missing captions for Figure 2. We have added captions in the revised paper.
>
>
> >**W2, Q1**: The analysis in Proposition 3.1 is limited to the tabular case and cannot be directly applied to humanoid whole-body control. The justification should be revised, possibly using covering number arguments for general function approximation.
>
> Thanks for the insightful suggestion. We agree that the original analysis of BC is limited to the tabular case and cannot be directly applied to humanoid whole-body control. To address this limitation and provide a more rigorous theoretical foundation, we derive the expert sample complexity of BC within the General Function Approximation (GFA) framework, following the conclusion established in [1]. In this context, the required expert sample complexity for BC is bounded by $\tilde{\mathcal{O}}(\frac{H^3\log\mathcal{N}(\Pi,\varepsilon_\pi)}{\varepsilon^2})$, where $H$ is the task horizon,  $\Pi$ is the policy class, and $\mathcal{N}(\Pi,\varepsilon_\pi)$ is the covering number of $\Pi$. We have revised the theoretical analysis section of the paper to include this derivation and explicitly state this complexity bound.
>
>
> >**W2, Q1**: The proposed hierarchical method decomposes the action space, not the state space, which may not align with the theoretical bound based solely on state-space cardinality.
>
> **TL;DR: We clarify that HVD improves the sample efficiency by (i) reducing the size of the policy class via prior injection and (ii) leveraging broader data sources.**
>
> Thank you for your valuable question. We posit that our method, HVD, improves sample efficiency from two perspectives, and from a theoretical standpoint, action decomposition also benefits policy performance:
>
> 1. **Reducing the covering number of the policy class.** Under the GFA analysis framework, the sample complexity of imitation learning depends on the covering number of the policy class, denoted as $\mathcal{N}(\Pi, \varepsilon_\pi)$. As the action space expands, $\mathcal{N}(\Pi, \varepsilon_\pi)$ grows, requiring more expert trajectories to learn an $\varepsilon_\pi$-optimal policy. By incorporating human priors through **action decomposition**, HVD effectively reduces the complexity of the policy class, which leads to improved sample efficiency.
> 2. **Leveraging broader data sources.** HVD further enhances generalization by enabling the policy to utilize suboptimal demonstrations via offline reinforcement learning. This allows the agent to learn not only from optimal expert data but also from imperfect or diverse trajectories, thereby improving robustness and overall policy performance.

---

> ### Author Response · Authors · 2025-11-21
>
> >**W3, Q2**: The authors should include more offline RL baselines based on generative policies that also utilize rewards for a fair evaluation.
>
> **TL;DR: We add QIPO [2] as an offline RL baseline and show that HVD consistently performs better than this baseline.**
>
> Thank you for your valuable suggestions. We reimplement QIPO based on $\pi_0$ and conduct it on multi-task learning setting. To ensure a fair comparison, QIPO and HVD share identical network architectures, optimizer settings, and training tricks; the only differences lie in (i) the policy update mechanism and (ii) the use of hierarchical value decomposition in HVD.
>
> Specifically, QIPO first initializes its Q-network using a combination of behavior cloning (BC) and temporal difference (TD) losses, followed by iterative policy updates. We follow the hyperparameter recommendations from the original QIPO paper as a starting point and perform minimal tuning to ensure stable convergence. The final configuration uses $\beta=1, M=16, K_\text{renew}=10$.
>
> The results are summarized in the table below. HVD consistently achieves higher success rates across all five tasks. The performance gap is especially pronounced in long-horizon, whole-body control tasks, where HVD may benefit from its more accurate credit assignment enabled by its hierarchical value decomposition.
>
> | **Method**    | Pen Insert | Cup Upright | Wipe Board | Basket Carry | Trash Dispose | **Avg SR** |
> |--------------------------|------------|-------------|------------|--------------|---------------|------------|
> | QIPO    | 0.90 | 0.90 | 0.30 | 0.20 | 0.25 | 0.51    |
> | HVD (ours) | **0.92**   | **0.94**    | **0.32**   | **0.50**     | **0.33**          | **0.60**   |
>
> >**W4**: Several relevant studies on humanoid whole-body control that also exploit hierarchical structures are missing from the discussion and should be cited for completeness.
>
> Thanks for pointing out this problem. We have updated the Related Work section in the revised paper to include prior studies on humanoid whole-body control that also employ hierarchical structures, such as [3, 4, 5]. Each of these approaches demonstrates how structuring control into hierarchical layers or skill modules can improve the scalability and robustness of humanoid behavior learning in long-horizon, high-DoF tasks. In contrast, HVD introduces hierarchy in the value learning process, decomposing Q-values along kinematic dimensions to enable more precise credit assignment and scalability in offline RL settings.
>
>
> ### References
> [1] Is Behavior Cloning All You Need? Understanding Horizon in Imitation Learning. NeurIPS’24.
>
> [2] Energy-Weighted Flow Matching for Offline Reinforcement Learning. ICLR'25.
>
> [3] Hierarchical World Models as Visual Whole-Body Humanoid Controllers. ICLR’25.
>
> [4] HMC: Learning Heterogeneous Meta-Control for Contact-Rich Loco-Manipulation. RSS’25.
>
> [5] HumanPlus: Humanoid Shadowing and Imitation from Humans. CoRL’24.

---

> > ### Comment · Reviewer_xcm5 · 2025-11-21
> >
> > Thank you for your reply. I have a follow-up question regarding the theoretical analysis. The new result by Foster et al. presented in your revised manuscript is for finite-horizon episodic MDPs. However, in your preliminaries, the MDP is defined in the discounted setting. Could you please translate or adapt the result so that it aligns with your discounted MDP formulation?
> >
> > Moreover, I recommend moving some of the comparisons with offline RL methods into the main text and restructuring the manuscript, as the current version exceeds the 9-page limit. I will consider increasing my score once these changes are implemented.
> >
> > Best regards,
> >
> > Reviewer

---

> > > ### Author Response · Authors · 2025-11-21
> > >
> > > Thank you for your careful review and insightful follow-up comments. We sincerely appreciate your feedback and have revised the paper accordingly:
> > >
> > > 1. We apologize for the ambiguity in the definition of the MDP in the Preliminaries section. To clarify, **our method and experiments are strictly based on the finite-horizon episodic MDP framework, with no discount factor involved.** We have updated the definition in the Preliminaries section to ensure it aligns with this framework and makes the referenced theoretical results directly applicable.
> > > 2. We have included additional offline RL comparison results in the main text, specifically in Section 5.5.
> > > 3. We note that, according to the ICLR 2025 author guidelines, **rebuttal-stage submissions can be up to 10 pages**. Our revised paper, with the clarified preliminaries and new comparison, remains within this limit.
> > >
> > > We would be pleased to provide any further clarification if needed. Thank you once again for your thoughtful feedback.

---

> > > > ### Comment · Reviewer_xcm5 · 2025-11-21
> > > >
> > > > Thanks for your clarification and updates. I think my concerns have been addressed. I'll be happy to raise my score.
> > > >
> > > > Best,
> > > >
> > > > Reviewer

---

> > > > > ### Author Response · Authors · 2025-11-28
> > > > >
> > > > > We are happy that all your concerns have been addressed. We sincerely appreciate the reviewer for raising the score.

---

### Official Review · Reviewer_ruzg · 2025-10-27

**Soundness:** 2
**Presentation:** 3
**Contribution:** 2
**Rating:** 4
**Confidence:** 2

**Summary:**

The paper indicates that imitation-learning approaches suffer from state-action-space explosion when applied to whole-body control, as a consequence it's unrealistic to collect the exponentially-growing expert data.
On the other hand, suboptimal trajectories are more readily available, motivating the use of offline reinforcement learning.
The paper proposes to use hierarchical value deomposition based on the embodiment (body, torso, arm) to improve credit assignment.
To demonstrate the improvement, the paper provides a curated dataset consists of optimal and suboptimal demonstrations, and policy rollouts, for multiple tasks such as pen insert, cup upright, wipe board, basket carry, and trash dispose.
On this dataet, the paper shows that hierarchical-value deomposition outperforms the standard imitation-learning approach on three vision-language-action models.

**Strengths:**

- The paper is clear and easy-to-follow.
- The experiments are conducted on a real-world robot that requires substantial efforts.

**Weaknesses:**

- Motivation
	- In the abstract, the paper states that scaling imitation learning (IL) to high DoF is limited by non-stationary observation transition. My concern with this statement is that if this is indeed true, the paper should not be focusing on formulating whole-body control as a Markov decision process (MDP), since MDPs do assume stationary transitions.
	- The paper indicates, on lines 38-42, that scaling IL approaches is fundamentally limited by the number of state-actions exploding exponentially. I agree with this statement, but I also agree that offline reinforcement learning (RL) approaches also suffer from this problem. In fact, one can argue that offline RL scales worse without data coverage and appropriate realizability assumptions [1].
- Method
	- Eq. 1: Why this choice of ordering the base, torso, and arm? Is there any ablation to demonstrate that this is the "best" way to decompose the Q-function?
	- Eq. 2: It seems like for the later Q-functions, we can simply regress based on, e.g.,  $Q_{base} - Q_{torso}$. That is, why not $Q(s, a) - Q(s, a, a')$? From one perspective we can view $Q_{base}$ as the baseline for $Q_{torso}$ and so on.
- Experiments
	- While I appreicate the challenges in producing results on a real robot, I strongly believe reproducibility is important---as a result, it would be great if the paper can provide even a single simulated environment, perhaps MuJoCo humanoid, that this approach can indeed improve upon existing methods.
		- I am unsure where the dataset, WB-50, will be provided---that will be greatly appreciated for reproducibility purposes as well.
	- What is the number of rollouts for evaluation?
	- How is "better credit assignment" analyzed? I was hoping for analysis on Q-values being more accurately propagated with the decomposition.
	- While the theory suggests the exponential explosion due to increasing DoF, it would be nice if empirically the paper demonstrates this problem with ablation on dataset size vs performance on increasing number of DoF.

References:
[1] Zhan, Wenhao, et al. "Offline reinforcement learning with realizability and single-policy concentrability." Conference on Learning Theory. PMLR, 2022.

**Questions:**

See weakness above

---

> ### Author Response · Authors · 2025-11-21
>
> We sincerely appreciate the reviewer’s constructive feedback and would like to offer further clarification in response.
>
> >**W1**: In the abstract, the paper states that scaling imitation learning (IL) to high DoF is limited by non-stationary observation transition. My concern with this statement is that if this is indeed true, the paper should not be focusing on formulating whole-body control as a Markov decision process (MDP), since MDPs do assume stationary transitions.
>
> **TL;DR: We explain why we formulate whole-body control as a non-stationary MDP and provide the basis for this modeling choice.**
>
> We appreciate the reviewer's valuable comment and would like to provide clarification. Because the **whole-body movement** and the **constraints of the camera's Field of View (FoV)** result in partial observability, **making the observation-based transitions appear highly stochastic and time-varying throughout the horizon.** As a result, the transition function in visual observation for robots may be **stochastic**, thus we assume it to be non-stationary.
>
> Additionally, non-stationary MDPs are commonly used in the RL and IL community, as demonstrated by works such as [1, 2, 3]. Following these works, we adopt a similar approach for defining our notation in the preliminary section. We revised this part of the manuscript to make the assumption of "non-stationarity" more explicit and clearly justified.
>
> >**W2**: The paper indicates that scaling IL approaches is fundamentally limited by the number of state-actions exploding exponentially. But I also agree that offline reinforcement learning (RL) approaches also suffer from this problem. In fact, one can argue that offline RL scales worse without data coverage and appropriate realizability assumptions.
>
> Thank you for the valuable comment. Indeed, offline RL faces this challenge.  However, even in cases where the data coverage is extremely narrow, **due to HVD combining both BC and offline RL terms (Equation 8)** during the training phase, ensuring that **the policy performance of HVD is no worse than BC**. Moreover, during daily data collection and testing, we generate a large volume of non-expert trajectories. IL, which relies on optimal expert data, cannot exploit these at all, whereas offline RL can leverage them to mitigate data scarcity, facilitating more efficient policy learning.
>
> >**W3**: Eq. 1: Why this choice of ordering the base, torso, and arm? Is there any ablation to demonstrate that this is the "best" way to decompose the Q-function?
>
> **TL;DR: We articulate the rationale behind the base–torso–arm decomposition order and provide empirical evidence demonstrating its clear superiority over alternative strategies.**
>
> Thank you for your insightful question. Our decoding order design is motivated by the following principles: First, from a control stability perspective, we observe that errors in the mobile base propagate significantly to the torso and arm, whereas the reverse effect is considerably less pronounced. This sequential decoding strategy effectively minimizes error propagation throughout the entire kinematic chain. Second, for task performance optimization, this hierarchy ensures that the arm, responsible for fine manipulation, operates on a more stable foundation, leading to greater precision and improved coordination among all components. Finally, this approach is consistent with human demonstrator behavior, as evidenced by our teleoperation data, which shows a similar execution pattern during whole-body tasks.
>
> Moreover, to validate the impact of different decomposition orders and validate our design choice, we conduct an ablation experiment. Specifically, we compare two alternative sequencing strategies: (i) Arm–Torso–Base and (ii) Base–Arm–Torso, using the same base policy $\pi_0$ across all tasks within a multi-task learning setup. Due to time constraints, the baseline policy is evaluated over 20 trials per task. The results are summarized in the Table below, **clearly demonstrating that our proposed decomposition order outperforms the alternatives,** highlighting the importance of the hierarchical structure in our approach.
>
> | **Method**               | Pen Insert | Cup Upright | Wipe Board | Basket Carry | Trash Dispose | **Avg SR** |
> |--------------------------|------------|-------------|------------|--------------|---------------|------------|
> | Arm-Torso-Base     | 0.55       | 0.60        | 0.15       | 0.20         | 0.10          | 0.32       |
> | Torso-Arm-Base    | 0.80       | 0.80        | 0.20       | 0.35         | 0.20          | 0.47       |
> | Base-Torso-Arm (ours) | **0.92**   | **0.94**    | **0.32**   | **0.50**     | **0.33**          | **0.60**   |

---

> ### Author Response · Authors · 2025-11-21
>
> >**W4**: Eq. 2: It seems like for the later Q-functions, we can simply regress based on, e.g., . That is, why not ? From one perspective we can view as the baseline for and so on.
>
> **TL;DR: We provide a concise analysis showing that residual Q-decomposition is a valid variant of HVD. We further validate this empirically and find that our proposed independent decomposition yields better performance than the residual variant.**
>
> Thank you for the valuable question. We agree that learning the residual between Q-functions at different hierarchies is also a concrete implementation for HVD. Concretely, we first define that $Q_\text{torso}=Q_\text{base} + \delta Q_\text{torso}$, where $Q_\text{base}$ and $\delta Q_\text{torso}$ are parameterized by neural networks. In such a context, the resulting TD loss becomes
> $\mathcal{L} (Q_\text{torso}) = \left( \delta Q_\text{torso} + (r + Q_\text{base} - Q^\prime_\text{base}) - \delta Q^\prime_\text{torso} \right)^2$. This can be viewed as reshaping the original fixed reward into a moving target $r^\prime = r + Q_\text{base} - Q^\prime_\text{base}$, which changes with every update of $Q_\text{base}$. This non-stationary regression objective introduces time-varying supervision and may destabilize training.
>
> To further validate the residual approach versus our proposed independent decomposition, we add an additional baseline across all tasks within a multi-task learning setup, which learns the residual Q-function $Q_{\text{residual}}$ using the formulation defined above. Due to time constraints, the baseline policy is evaluated over 20 trials per task. The results are as below, **showing that our independent decomposition outperforms the residual variant baseline, achieving a higher average success rate.**
>
> | **Method**    | Pen Insert | Cup Upright | Wipe Board | Basket Carry | Trash Dispose | **Avg SR** |
> |--------------------------|------------|-------------|------------|--------------|---------------|------------|
> | HVD (residual)     | 0.75 | 0.80 | 0.25 | **0.50** | **0.35** | 0.53    |
> | HVD (ours) | **0.92**   | **0.94**    | **0.32**   | **0.50**     | 0.33          | **0.60**   |
>
>
>
> >**W5**: It would be great if the paper can provide even a single simulated environment.
>
> **TL;DR: We conduct whole-body control experiments on the BEHAVIOR-1K simulation benchmark and demonstrate that HVD significantly improves performance over the IL baseline.**
>
> We thank the reviewer for raising this point. We add a simulation experiment on the BEHAVIOR-1K platform [4], a widely used benchmark for whole-body control. We select the Picking Up Trash task and decompose it into two core evaluation stages: (i) Approach to the trash can,  (ii) Grasp the trash can, to focus on the critical whole-body coordination and high-dimensional action challenges.
>
> We train the base policy $\pi_0$ using a mixed dataset of 300 trajectories:  200 expert demonstrations from the official BEHAVIOR-1K dataset and 100 rollouts collected from a pretrained $\pi_0$ baseline. HVD is implemented directly on top of the official $\pi_0$ codebase, ensuring fair comparison. We use binary success labels as rewards, applying 1 for successful trajectories and 0 for failures, provided at the end of each episode. Additionally, every intermediate frame incurs a small step penalty of −0.001 to encourage efficiency. During evaluation, we test each policy on 100 trials across 20 unseen scenarios. Full details of the HVD architecture and training process are consistent with the description in **Appendix D.4** in the revised paper.
>
> The results are shown below. Our results demonstrate that HVD significantly outperforms the IL baseline. HVD achieves a 77.0% success rate in stage 1, improving upon 45.0% of IL. The gains are clearer in the more complex Grasping stage, **where HVD has 22.0% success versus 5.0% of IL.**
>
> | Method       | Stage 1 (SR) | Stage 2 (SR) |
> |--------------|------------|------------|
> | $\pi_0$-IL   | 45.0%      | 5.0%      |
> | $\pi_0$-HVD  | **77.0%**      | **22.0%**      |
>
>
> >**W6**: I am unsure where the dataset, WB-50, will be provided---that will be greatly appreciated for reproducibility purposes as well.
>
> We will release WB-50 upon publication of this paper.
>
> >**W7**: What is the number of rollouts for evaluation?
>
> Thank you for pointing out this issue. For evaluation, **each policy was tested with 50 independent rollouts per task,** using randomized initializations of robot pose, object placement, and background perturbations as described in **Section 5.1**. The reported success rates and stage scores represent the average performance across these rollouts. All baseline methods and HVD were evaluated under the same protocol to ensure fairness and statistical reliability. We have added the rollout details to the revised manuscript.

---

> ### Author Response · Authors · 2025-11-21
>
> >**W8**: How is "better credit assignment" analyzed? I was hoping for analysis on Q-values being more accurately propagated with the decomposition.
>
> **TL;DR: We conduct a detailed credit assignment analysis across three tasks by examining the hierarchical weights learned by HVD. The results show that HVD successfully and interpretably allocates higher weights to the robot components most critical for success during specific task stages.**
>
> Thank you for raising this important point. To thoroughly investigate how our method enables more accurate credit assignment, we conducted an experimental analysis by **extracting the hierarchical weights on the training dataset and computing their average values across different task stages.** The results are summarized in the table below.
>
> | Task           | Stage                                      | Base   | Torso  | Arm    |
> |----------------|--------------------------------------------|--------|--------|--------|
> | Wipe Board     | Stage 1: Approach Whiteboard and Grasp Eraser | 0.962  | 0.884  | 1.321  |
> |                | Stage 2: Wipe Markings                      | 0.946  | 0.890  | 1.054  |
> |                | Stage 3: Return Eraser                      | 0.857  | 0.826  | 1.149  |
> | Basket Carry   | Stage 1: Approach Basket                    | 0.980  | 0.999  | 1.405  |
> |                | Stage 2: Lift Basket                        | 0.953  | 0.909  | 1.254  |
> |                | Stage 3: Place Basket on Table              | 0.837  | 0.807  | 1.063  |
> |                | Stage 4: Place Markers in Basket            | 0.972  | 0.980  | 1.243  |
> | Trash Dispose  | Stage 1: Open Trash Can Lid                 | 0.933  | 0.918  | 1.382  |
> |                | Stage 2: Grasp Paper Towel                  | 0.930  | 0.960  | 1.273  |
> |                | Stage 3: Dispose of Paper Towel             | 0.944  | 0.902  | 1.356  |
> |                | Stage 4: Close and Lock Trash Can Lid       | 0.804  | 0.813  | 1.323  |
>
>
> Our analysis reveals that the hierarchical weights effectively capture the varying importance of different robot components (base, torso, arm) throughout task execution. In the Wipe Board task, for instance:
>
> 1. In Stage 1 (Approach Whiteboard and Grasp Eraser), the model assigns higher weights to both the base and arm, indicating its recognition that navigation and precise grasping are crucial for successfully approaching the whiteboard and picking up the eraser.
> 2. In Stage 2 (Wipe Markings), the arm maintains the highest weight as the robot wipes the board, while the base and torso weights remain relatively stable. Notably, the torso weight shows a slight increase, potentially reflecting the need to maintain balance when applying force against the whiteboard.
> 3. For Stage 3 (Return Eraser), the arm receives significantly higher weight as precise placement of the eraser back into its slot becomes critical, while both base and torso weights decrease accordingly.
>
> Furthermore, we observe that during delicate manipulation phases (e.g., picking up and placing the eraser in Stages 1 and 3), the arm is consistently assigned higher weights compared to Stage 2, highlighting the model's ability to adapt credit assignment based on precision requirements.
>
> Similar stage-aware credit assignment patterns are evident in other tasks (Basket Carry and Trash Dispose), as detailed in the table. These results collectively demonstrate that our approach successfully differentiates between task stages and allocates appropriate weights to different components, thereby enabling more accurate and interpretable credit assignment throughout the hierarchical decision-making process. We add this analysis in **Appendix D.3** of the revised paper.

---

> ### Author Response · Authors · 2025-11-21
>
> >**W9**: While the theory suggests the exponential explosion due to increasing DoF, it would be nice if empirically the paper demonstrates this problem with ablation on dataset size interacts with performance on increasing number of DoF.
>
> **TL;DR: We clarify the gap between worst-case theoretical bounds and real-world performance, refine our analysis using the GFA framework to better reflect practical settings, and add a new ablation study on dataset size vs performance on increasing number of DoF.**
>
> Thank you for the insightful suggestion. First, we would like to clarify that our original theoretical analysis was based on a **worst-case, tabular setting.** In such a setting, the exponential dependence on the DoF arises under highly pessimistic assumptions that are rarely satisfied in real-world robotic systems. **The goal of this analysis was not to claim that this exponential scaling occurs in practice**, but rather to theoretically highlight how increasing the action dimensionality can drastically increase the demand for expert data.
>
> To better bridge theory and practice, **we now frame expert data requirements within the General Function Approximation (GFA) framework.** Deriving from [5], the expert complexity of Behavior Cloning (BC) in GFA is $\tilde{\mathcal{O}}(\frac{H^3\log\mathcal{N}(\Pi,\varepsilon_\pi)}{\varepsilon^2})$, where $H$ is the task horizon,  $\Pi$ is the policy class, and $\mathcal{N}(\Pi,\varepsilon_\pi)$ is the covering number of $\Pi$. Therefore, as the action space expands, $\mathcal{N}(\Pi, \varepsilon_\pi)$ grows, requiring more expert trajectories to learn an $\varepsilon_\pi$-optimal policy. We provide a detailed discussion of this refined analysis in **Section 3.1** of the revised paper.
>
> As an initial demonstration, our empirical study in **Section 3.2 (Figure 1, rightmost panel)** shows that under a fixed dataset size of 50 expert demonstrations, the success rate drops substantially when moving from arm-only control to full-body control. Despite identical data size and evaluation protocol, the 21-DoF setting consistently yields lower success rates, confirming that higher action dimensionality leads to increased sample complexity in practice.
>
> To further support this claim, we conduct an additional experiment where we varied the number of expert trajectories (25, 50, 100) and action space dimensionality (7-DoF vs 21-DoF). As shown in the table below, the success rates of baseline method consistently improve with more data, but the performance gap between low-DoF and high-DoF settings persists. **This empirical results highlight that higher action dimensionality imposes steeper expert data requirements.**
>
> |      Data Size    | 25  | 50  |  100 |
> |-----------------|--------------------------|--------------------------|--------------------------|
> | $\pi_0$ (7-DoF)      | 0.48      | 0.88        | 0.94         |
> | $\pi_0$ (21-DoF)     | 0.28      | 0.46        | 0.62         |
>
> ### References
> [1] Provably Efficient Reinforcement Learning with Linear Function Approximation. COLT’20.
>
> [2] Toward the Fundamental Limits of Imitation Learning. NeurIPS’20.
>
> [3] Provably Efficient Adversarial Imitation Learning with Unknown Transitions. UAI’23.
>
> [4] BEHAVIOR-1K: A Human-Centered, Embodied AI Benchmark with 1, 000 Everyday Activities and Realistic Simulation.
>
> [5] Is Behavior Cloning All You Need? Understanding Horizon in Imitation Learning. NeurIPS’24.

---

> > ### Comment · Reviewer_ruzg · 2025-11-23
> >
> > Thank you for the detailed response. Overall I'm happy with the rebuttal and have increased the score accordingly.
> >
> > Regarding MDP---yes I agree, though my point is that non-stationarity can come from the perspective of partial observability, to that end even with time-dependent transition function it's insufficient to model this (without blowing up the state space even more).

---

> ### Author Response · Authors · 2025-11-24
>
> Thank you for your thoughtful reply and insightful comment.
>
> We agree that partial observability poses a challenge in whole-body control, and that modeling the environment solely as a non-stationary MDP may fall short of capturing the underlying dynamics. Incorporating a Partially Observable Markov Decision Process (POMDP) framework, or other methods designed to handle partial observability, could offer a more principled approach. This is a compelling direction we hope to explore in future work.
>
> Thank you again for your valuable feedback!

---

### Official Review · Reviewer_87vt · 2025-10-29

**Soundness:** 3
**Presentation:** 3
**Contribution:** 3
**Rating:** 6
**Confidence:** 4

**Summary:**

The authors aim to apply offline RL methods to whole-body robotic control. In general, whole-body control is more complicated than only manipulation control, due to larger degrees of freedom, increased variation from adjusting robot position, etc. These increase the complexity of the RL problem.

To address this, the authors propose using a hierarchical offline RL approach. Instead of treating the action $a$ as a monolithic action, it is segmented into three components, $a_{base}, a_{torso}, a_{arm}$. A Q-function is trained with each conditioning on all prior modalities, giving 3 Q-functions $Q(s, a_{base})$, $Q(s, a_{base}, a_{torso})$, and $Q(s, a_{base}, a_{torso}, a_{arm})$. Each is fit with its own TD loss term  but shares base architecture. For learning the Q-function and extracting a policy at the end, the authors use IDQL from prior work (fit Q-function with TD-learning then apply diffusion AWR to extract a policy). Each action component $a_{base}, a_{torso}, a_{arm}$ is extracted using advantage weights computed from the corresponding Q-function.

To evaluate this, the authors collect a 50 hour dataset of whole body control on the Galaxea platform, made of a mix of expert demonstrations, suboptimal demonstrations from operators that are worse at teleoperation, and policy rollouts from learned policies. The authors show their offline RL outperforms pure IL, and that the multi-level Q-function performs better than using no hierarchy.

**Strengths:**

The paper tackles an important problem of learning from suboptimal offline data. Collecting and labeling a 50 hour dataset is potentially quite useful. The paper acknowledges some of the weaknesses of the approach as well (specifically the need to do this labeling). Ablations suggest there are gains from doing this more hierarchical setup, and visualization of advantage weights over the subtask are compelling.

**Weaknesses:**

The theoretical analysis seems entirely unnecessary to me. The cited Rajaraman et al 2020 bound is based on a worst case assumption of environment dynamics and generalization, and given that empirical dataset sizes are significantly smaller than this bound, it doesn't really seem to apply and just seems like an overly complicated way to state that higher dimensionality problems may need more data to fit.

The specific decomposition into base, torso, and arm is pretty specific to whole-body control. It seems like a special case of autoregressively fitting + extracting a Q-function 1 action dimension at a time, similar to Q-Transformer (although Q-transformer does not do any AWR weighting / diffusion step).

It was unclear to me how many trials were used to generate the task success rate tables of Table 1 / Table 2, or how much the hierarchical setup affects action inference time.

**Questions:**

Are expert demonstrations vs suboptimal rollouts evenly split among tasks, or are some tasks more biased towards one or the other?

How is inference speed affected by dividing the action space this way?

Why is so much of the appendix spent defining quantitative measures of task complexity that don't seem to be used anywhere else?

---

> ### Author Response · Authors · 2025-11-21
>
> We sincerely appreciate the reviewer’s constructive feedback and would like to offer further clarification in response.
>
> >**W1**: The theoretical analysis seems entirely unnecessary to me, it doesn't really seem to apply and just seems like an overly complicated way to state that higher dimensionality problems may need more data to fit.
>
> **TL;DR: We reframe the theory under the GFA framework to show that BC’s expert sample complexity grows with action-space dimensionality, and explain why HVD’s structured decomposition improves sample efficiency.**
>
> Thank you for the valuable comment. In the original version, rather than pinning down an exact number of expert trajectories needed to solve each task, we use the theory here to **highlight that the required amount of expert data grows linearly with the size of the state space, and that size itself expands exponentially with its dimension.** Consequently, the high-DoF nature of whole-body control explodes sample complexity.
>
> To better bridge theory and practice, we now frame expert data requirements within the General Function Approximation (GFA) framework. Deriving from [1], the expert complexity of Behavior Cloning (BC) in GFA is $\tilde{\mathcal{O}}(\frac{H^3\log\mathcal{N}(\Pi,\varepsilon_\pi)}{\varepsilon^2})$, where $H$ is the task horizon,  $\Pi$ is the policy class, and $\mathcal{N}(\Pi,\varepsilon_\pi)$ is the covering number of $\Pi$. Therefore, as the action space expands, $\mathcal{N}(\Pi, \varepsilon_\pi)$ grows, requiring more expert trajectories to learn an $\varepsilon_\pi$-optimal policy.
>
> **HVD addresses this issue by incorporating human priors through action decomposition and leveraging suboptimal data, which restricts the policy class to a more structured and lower-complexity hypothesis space.** This reduction in $\mathcal{N}(\Pi, \varepsilon_\pi)$ directly improves sample efficiency. We have updated the theoretical analysis in the revised paper.
>
> >**W2**: The specific decomposition into base, torso, and arm is pretty specific to whole-body control. It seems like a special case of autoregressively fitting + extracting a Q-function 1 action dimension at a time, similar to Q-Transformer.
>
> **TL;DR: We clarify key distinctions between HVD and Q-Transformer along three dimensions: (1) the problem setting, (2) the structure of Q-function decomposition, and (3) the policy extraction mechanism.**
>
> We appreciate the reviewer's valuable comment and would like to provide clarification. We agree that both HVD and Q-Transformer are used for scalable offline RL, but the two methods differ fundamentally in (1) the problem they tackle and (2) the methodological role the factorization plays. Below, we clarify why the HVD decomposition is **not** simply a special case of Q-Transformer's autoregressive approach:
>
> 1. **Different problems addressed.** Q-Transformer seeks to answer the question, "How can we integrate offline Q-learning into a Transformer model?" In contrast, HVD addresses the problem of "How can we enable **existing** **whole-body** VLA/diffusion policies to learn from **suboptimal data**?"
> 2. **Different Q-function decomposition.** Q-Transformer decodes the Q-function in a **one-dimensional-at-a-time** manner, whereas HVD decodes it **one-hierarchy-at-a-time**. Additionally, while Q-Transformer only test their effectiveness on image and text modalities, HVD is capable of processing 3D point cloud data, thereby extending its applicability.
> 3. **Different policy extraction methods.** Q-Transformer extracts actions by applying an argmax to a discretized Q-function, which restricts its flexibility when adapting to pre-trained VLA policies. In contrast, HVD adapts to these policies via AWR weighting, making it compatible with most diffusion-based or flow-matching VLA models. As such, HVD can be seen as a post-training method that enhances the flexibility and utility of VLA policies.
>
> > **W3**: It was unclear to me how many trials were used to generate the task success rate tables of Table 1 / Table 2.
>
> We apologize for not clearly specifying the number of trajectories used for testing. For all experiments in Table 1 and Table 2, we conducted **50 evaluation trials per task per method** to ensure statistical reliability of the reported success rates.
>
> > **W3, Q2**: How much the hierarchical setup affect action inference time.
>
> Thank you for the valuable question. **Our hierarchical framework introduces no additional overhead during policy inference.** The hierarchical decomposition is applied only to the Q-function during training, while the policy network architecture remains unchanged. Specifically, we first train a Q-network , then use it to compute adaptive weights for policy learning without any modification of the policy network architecture, maintaining identical inference speed. This design ensures that our approach achieves improved performance without sacrificing inference efficiency.

---

> ### Author Response · Authors · 2025-11-21
>
> >**Q1**: Are expert demonstrations vs suboptimal rollouts evenly split among tasks, or are some tasks more biased towards one or the other?
>
> **TL;DR: We clarify that the data distribution is roughly balanced across all tasks and list the exact proportions for each task. We further empirically validate that HVD is robust to varying data distribution.**
>
> Thank you for the valuable question. The proportion of expert and suboptimal trajectories is approximately balanced across all five tasks in the WB-50 dataset. We did not artificially adjust the data distribution for each task. Instead, we simply collected all trajectories generated during human teleoperation and policy rollouts, preserving their natural distribution. Specifically, for each task, the ratio of expert demonstrations to suboptimal trajectories (suboptimal human data and policy rollouts) varies within $\pm$ 5.5% of the overall dataset composition (expert 43.7%, suboptimal 56.3%). The detailed per-task proportions of expert and suboptimal trajectories are summarized below. For the proportion of frames per task, please refer to the **Figure 20 in Appendix B.7.**
> | Task              | Pen Insert | Cup Upright | Wipe Board | Basket Carry | Trash Dispose |
> |-------------------|------------|-------------|------------|--------------|----------------|
> | Expert    | 38.2%    | 38.8%    | 45.0%       | 49.2%         | 40.7%           |
> | Suboptimal | 61.8%   | 61.2%     | 55.0%       | 50.8%         | 59.3%           |
>
> To further validate the data distribution robustness of HVD, we conduct an additional ablation on the Pen Insert task under varying expert ratios. During training, we keep the total number of demonstrations fixed at 100 with expert ratios of 20%, 50%, and 80%. During training, $\pi_0$+IL uses only the expert demonstrations, while $\pi_0$+HVD uses all demonstrations, including both expert and suboptimal data. The results are shown below, where **HVD consistently outperforms IL, confirming that HVD maintains its effectiveness across different data compositions.** Due to time constraints, each policy is evaluated over 20 trials.
>
> | Method          | 20 exp + 80 imp | 50 exp + 50 imp | 80 exp + 20 imp |
> |-----------------|--------------------------|--------------------------|--------------------------|
> | $\pi_0$+IL      | 0.15      | 0.35        | 0.50          |
> | $\pi_0$+HVD     | **0.55**  | **0.60**     | **0.75** |
>
>
>
> >**Q3**: Why is so much of the appendix spent defining quantitative measures of task complexity that don't seem to be used anywhere else?
>
> **TL;DR: The quantitative measures of task complexity are provided to assess task difficulty, identify baseline bottlenecks, and explain the effectiveness of the HVD approach. We also provide the source of these measures.**
>
> Thank you for the insightful comment. The task complexity metrics presented in Appendix B are not used as inputs to the learning algorithm. Instead, their purpose is to **quantitatively characterize the spectrum of task difficulty** across our real-world benchmark suite. **By correlating these difficulty metrics with the actual success rates of learned policies, we can identify key bottlenecks in current whole-body control approaches and pinpoint the primary sources of performance gains introduced by HVD.**
>
> For example, by correlating the difficulty metrics with the actual success rates of learned policies, we can identify that high control complexity and high kinematic coordination demands are the primary failure modes for standard IL, which HVD successfully mitigates. We have expanded our discussion of these points in the **Appendix D.3** of the revised paper.
>
> This practice is common in recent robotic benchmarks. RLBench [2], for example, explicitly orders 100 manipulation tasks from simple goal-reaching to long-horizon tasks such as opening an oven and placing a tray inside. In the same spirit, our $C_{\text {time }}$ and $C_{\text {kinematic }}$ capture horizon and motion magnitude, while $C_{\text {control }}$ follows prior work that uses jerk-based measures to assess motion smoothness and control difficulty [3]. Finally, $C_{\text {coord }}$ measures how many joints are actively involved, motivated by works like RoboEval [4], which highlight that multi-arm, multi-stage, whole-body tasks are substantially more complex and demand higher coordination than single-step or single-arm tasks. Our metrics align with existing studies and serve to objectively document the difficulty range of our task suite, providing a reference for future work aiming to analyze algorithmic performance as a function of task complexity.
>
> ### References
>
> [1] Is Behavior Cloning All You Need? Understanding Horizon in Imitation Learning. NeurIPS’24.
>
> [2] RLBench: The Robot Learning Benchmark & Learning Environment. RAL.
>
> [3] Evaluating Uncertainty and Quality of Visual Language Action-enabled Robots.
>
> [4] RoboEval: Where Robotic Manipulation Meets Structured and Scalable Evaluation.

---

### Official Review · Reviewer_qv84 · 2025-11-01

**Soundness:** 4
**Presentation:** 4
**Contribution:** 3
**Rating:** 8
**Confidence:** 2

**Summary:**

This paper proposes Hierarchical Value-Decomposed (HVD) Offline Reinforcement Learning, a framework designed for high-dimensional, whole-body control in robotic systems. The authors motivate their approach through empirical and theoretical challenges in whole-body control. The authors then introduce the value decomposition framework, utilized within a transformer architecture, and achieve more accurate credit assignment across long-horizon behaviors. The authors present numerous results that show HVD significantly improves imitation performance, credit assignment, and can scale to multi-task settings.

**Strengths:**

+ Paper presents ample contributions, including a novel offline RL method for whole-body control, a dataset of whole-body behavior, and an implementation that support multi-modal and multi-task learning.
+ Paper is very well-written.

**Weaknesses:**

- The authors note that HVD relies on human-annotated rewards. Could you provide further information on where these come from and whether these are noisy reward signals? Further, does the framework depend on a specific distribution of data? For example, if 80% were imperfect demonstrations, would the framework still be able to learn behaviors well?

**Questions:**

Please address the weakness noted above.

---

> ### Author Response · Authors · 2025-11-21
>
> We sincerely appreciate the reviewer’s constructive feedback and would like to offer further clarification in response.
>
> >**W1**: The authors note that HVD relies on human-annotated rewards. Could you provide further information on where these come from and whether these are noisy reward signals?
>
> **TL;DR: We describe the details of the reward annotation pipeline, discuss noise arising from imperfect trajectory segmentation and sparsity of reward labels, and propose using multi-round human verification and reward chunking to mitigate these issues.**
>
> Thank you for valuable comments. In our work, human-annotated rewards follow the annotation protocol established in prior studies [1, 2]. Specifically, each demonstration trajectory is first manually segmented into key frames corresponding to individual sub-tasks. At each key frame, a reward label of 0.0 (failure), 0.5 (partial success), or 1.0 (full success) is assigned based on the degree of sub-task completion. Additionally, every intermediate frame incurs a small step penalty of −0.001 to encourage efficiency. Full details of the reward labeling procedure are provided in **Appendix B**.
>
> Moreover, the reward signals may be noisy, primarily due to two sources:
> 1. **Imperfect trajectory segmentation.** For continuous or fine-grained tasks, it is challenging for human annotators to consistently find the exact frame where a critical event occurs (e.g., the precise moment the gripper fully grasps the object or the tool makes contact), especially in high-frequency video streams.
> 2. **Sparsity of reward labels.** The discrete {0.0, 0.5, 1.0} signal provides only coarse-grained feedback and fails to capture the full continuum of task progress between key frames.
>
> To address these issues, several solutions are applied in our method:
> 1. **Multi-turn human verification.** Every annotated trajectory is independently reviewed by a second annotator to correct segmentation errors and label inconsistencies.
> 2. **Reward chunking.** We adopt the reward chunking technique [3], where the reward for a sequence of actions is defined as $r(s_t,a_{t:t+K})=\sum_{k=1}^K r(s_{t+k},a_{t+k})$. This formulation aggregates rewards over short action sequences, effectively smoothing out label noise and providing denser learning signals during training.
>
> Despite these imperfections, we found that this reward scheme substantially improves policy performance in our experiments, and we remain open to better labelling strategies in the future.
>
> >**W1:** Does the framework depend on a specific distribution of data? For example, if 80% were imperfect demonstrations, would the framework still be able to learn behaviors well?
>
> **TL;DR: We empirically validate that HVD is robust to varying data distribution.**
>
> Thank you for the insightful question. HVD does not assume any specific data distribution. Like prior offline RL methods, it leverages suboptimal demonstrations by learning a value function to reweight trajectories: high-quality segments receive higher imitation weights, while noisy segments are downweighted. Our hierarchical value decomposition technique further refines this process by decomposing the value function across different robotic components at each timestep.
>
> To empirically validate this robustness, we conduct an additional ablation on the Pen Insert task under varying expert ratios. During training, we keep the total number of demonstrations fixed at 100 with expert ratios of 20%, 50%, and 80%. During training, $\pi_0$+IL uses only the expert demonstrations, while $\pi_0$+HVD uses all available demonstrations, including both expert and suboptimal data. **The results are shown below, where HVD consistently outperforms IL, confirming that HVD maintains its effectiveness across a wide spectrum of data compositions.** Due to time constraints, each policy is evaluated over 20 trials.
>
> | Data           | 20 exp + 80 imp | 50 exp + 50 imp | 80 exp + 20 imp |
> |-----------------|--------------------------|--------------------------|--------------------------|
> | $\pi_0$+IL      | 0.15      | 0.35        | 0.50          |
> | $\pi_0$+HVD     | **0.55**  | **0.60**     | **0.75** |
>
>
> ### References
> [1] A Workflow for Offline Model-Free Robotic Reinforcement Learning. CoRL’21.
>
> [2] ConRFT: A Reinforced Fine-tuning Method for VLA Models via Consistency Policy. RSS’25.
>
> [3] Reinforcement Learning with Action Chunking. NeurIPS'25.

---

> > ### Comment · Reviewer_qv84 · 2025-11-25
> >
> > Thank you for your response. I will maintain my score of accept.

---

> ### Author Response · Authors · 2025-11-28
>
> We are happy that all your concerns have been addressed. We thank the reviewer for the recommendation for acceptance.

---

### Author Response · Authors · 2025-11-30
**Follow-up Summary for the Area Chair**

Dear AC,

We sincerely appreciate your tremendous efforts and valuable time in handling our submission. We are truly grateful for your guidance throughout this evaluation cycle. In our rebuttal and revised manuscript, we have thoroughly addressed all reviewer concerns, resulting in substantial improvements to the theoretical foundation, empirical evaluation, and methodological clarity of our work. Below is a concise summary of the key revisions:

- **Refined theoretical foundation (Section 3.1)**: We replace the original tabular analysis with a Generalized Function Approximation (GFA) framework that better aligns with real-world experimental settings, yielding a sample complexity bound of $\tilde{\mathcal{O}}(\frac{H^3\log\mathcal{N}(\Pi,\varepsilon_\pi)}{\varepsilon^2})$.
- **Clarifications on practical implementation (Section 5.1)**: We provide details on inference-time efficiency, commit to releasing the WB-50 dataset upon publication, and clarify evaluation protocols.
- **Expanded offline RL baselines (Section 5.5)**: We add QIPO as an offline RL baseline and show that HVD consistently performs better than this baseline.
- **Quantified credit assignment (Appendix D.2)**: We add empirical analysis showing HVD dynamically assigns higher weights to critical components per task stage, validating more accurate, interpretable credit propagation through decomposition.
- **Analysis of task complexity and policy performance (Appendix D.3)**: We clarify the purpose of task complexity measurement, identify baseline bottlenecks, and explain the effectiveness of the HVD approach.
- **New simulation experiments on BEHAVIOR-1K (Appendix D.4)**: We conduct whole-body control experiments on the BEHAVIOR-1K simulation benchmark and demonstrate that HVD significantly improves performance over the IL baseline.
- **Robustness across data distributions (Appendix D.5)**: We evaluate HVD under varying ratios of expert to suboptimal data (20%/50%/80%) and demonstrate consistent superiority over IL.
- **Analysis of decomposition order (Appendix D.6)**: We articulate the rationale behind the base–torso–arm decomposition order and provide empirical evidence demonstrating its clear superiority over alternative decomposition orders.
- **Comparison with residual Q-decoding (Appendix D.7)**:  We validate residual Q-decoding empirically and find that our proposed independent decomposition yields better performance than the residual variant.

---

Overall, we believe the revised paper now provides comprehensive evidence, resolves all reviewer concerns, and fairly reflects the contribution of our work. **Notably, Reviewer ruzg and Reviewer xcm5, who initially assigned negative scores, have confirmed that their concerns have been addressed and have raised their scores to 6 as of November 24, 2025. Furthermore, Reviewer qv84 has maintained an acceptance score of 8 after reviewing our rebuttal.**

We would like to express our gratitude once again for your dedicated handling of our paper. We sincerely appreciate the time and careful consideration you’ve devoted to reviewing our work.

Best regards,

The Authors

---

### Meta-Review · Area_Chair_LM3G · 2025-12-31

**Summary:**

This paper presents an interesting solution to an important problem. The author response provides a number of new experiments, revised theory, and additional analysis, improving the paper and addressing reviewer concerns. I recommend accept.

**Reviewer Concerns:**

I believe that most of the reviewer concerns are addressed by the rebuttal.

**Reviewer Scores:**

I believe that at least two of the reviewers would have increased their score.

---

### Decision · Program_Chairs · 2026-01-26

Accept (Poster)